# Nuclear numbers in syncytial muscle fibers promote size but limit the development of larger myonuclear domains

Alyssa A. W. Cramer [1], Vikram Prasad[1], Einar Eftestøl [1,2], Taejeong Song[3], Kenth-Arne Hansson[2,4], Hannah F. Dugdale[5], Sakthivel Sadayappan [3], Julien Ochala [5,6,7], Kristian Gundersen[2] & Douglas P. Millay [1,8 ✉]

Mammalian cells exhibit remarkable diversity in cell size, but the factors that regulate establishment and maintenance of these sizes remain poorly understood. This is especially true for skeletal muscle, comprised of syncytial myofibers that each accrue hundreds of nuclei during development. Here, we directly explore the assumed causal relationship between multinucleation and establishment of normal size through titration of myonuclear numbers during mouse neonatal development. Three independent mouse models, where myonuclear numbers were reduced by 75, 55, or 25%, led to the discovery that myonuclei possess a reserve capacity to support larger functional cytoplasmic volumes in developing myofibers. Surprisingly, the results revealed an inverse relationship between nuclei numbers and reserve capacity. We propose that as myonuclear numbers increase, the range of transcriptional return on a per nuclear basis in myofibers diminishes, which accounts for both the absolute reliance developing myofibers have on nuclear accrual to establish size, and the limits of adaptability in adult skeletal muscle.

[1] Division of Molecular Cardiovascular Biology, Cincinnati Children's Hospital Medical Center, Cincinnati, OH 45229, USA. [2] Department of Biosciences, University of Oslo, Oslo, Norway. [3] Department of Internal Medicine, Division of Cardiovascular Health and Disease, University of Cincinnati College of Medicine, Cincinnati, OH 45229, USA. [4] Center for Integrative Neuroplasticity (CINPLA), Department of Biosciences, University of Oslo, Oslo, Norway. [5] Center of Human and Applied Physiological Sciences, School of Basic and Medical Biosciences, Faculty of Life Sciences & Medicine, King's College London, London, UK. [6] Randall Center for Cell and Molecular Biophysics, School of Basic & Medical Biosciences, Faculty of Life Sciences & Medicine, Guy's Campus, King's College London, London, UK. [7] Department of Biomedical Sciences, University of Copenhagen, Copenhagen, Denmark. [8] Department of Pediatrics, University of Cincinnati College of Medicine, Cincinnati, OH 45229, USA. ✉email: douglas.millay@cchmc.org

The control of cell size is an evolutionarily conserved, fundamental aspect of development that must be highly regulated[1]. Cell size varies widely based on the type of cell and overall function, which involves a balance between available DNA and maintenance of biosynthetic processes for optimal structure and function[2,3]. Control of size in large cells, such as neurons and skeletal muscle fibers, requires unique adaptations to reach sufficient levels of transcripts and proteins throughout the large cytoplasm. These large cells are maintained in different ways, where neurons actively regulate transport of macromolecules produced in the cell body to the termini, while skeletal myofibers contain multiple nuclei dispersed throughout one cell[4,5]. A classic explanation for why muscle requires multiple nuclei is that each nucleus is capable of controlling only a certain volume of the cytoplasm, referred to as the myonuclear domain[6–12]. This thinking is consistent with the finding that exceeding the capability of a nucleus to sustain a given cytoplasmic volume disrupts cellular processes in organisms from yeast to mammalian cells[2,13,14]. While the number of myonuclei positively correlates with cell size in mice and humans, the threshold of myonuclei needed to achieve target cell size and whether there is flexibility for each of these myonuclei to maintain their cytoplasmic domains has not been adequately explored.

Multinucleated skeletal muscle fibers form from the fusion of progenitors during embryonic and postnatal development, and this fusion process is required for functional skeletal muscle. Indeed, genetic deletion of the muscle-specific fusion factors Myomaker and Myomerger in the mouse results in death just after birth[15–18]. In the adult mouse, a typical myofiber contains hundreds of nuclei, and possesses the ability to add new myonuclei in response to environmental cues owing to the presence of skeletal muscle stem cells[19]. There is robust myonuclear accretion in the mouse after birth, which slows around postnatal (P) day 21[20–22]. During this early time frame from P0 to P21, hindlimb myofibers grow in length and girth, although there is a greater increase in length compared to girth. P21 to adult is characterized by a greater radial growth compared to length[20,23], which we define as maturational growth. Here, since myofiber volume increases and the majority of myonuclear accretion occurs prior to the time of maturational growth, cytoplasm:DNA ratios increase indicating that the myonuclear domain is flexible during this postnatal period. Additional evidence for context-dependent myonuclear domains include that slow type I skeletal muscle fibers exhibit smaller size but increased myonuclear number compared to fast type II myofibers, which are larger with fewer myonuclei[24,25]. Given that skeletal muscle stem cells provide a mechanism for myofibers to accrue new nuclei through cellular fusion, increases in the cytoplasmic volume of adult myofibers can be achieved by regulating resident myonuclei, adding new myonuclei, or both. The majority of evidence indicates new myonuclei are required for functional adaptations, including hypertrophy in the adult, suggesting limited flexibility of resident myonuclei in adult myofibers[26–31]. Taken together, the decision for muscle to either add a new myonucleus or regulate nuclei that are already present in the myofiber is unclear and there is minimal understanding about the requirements for the full complement of resident myonuclei to establish functional cytoplasmic volumes during maturational growth.

In general, reasons for why large cytoplasmic:DNA ratios are challenging for cells to maintain include a need to support production and transport of sufficient synthetic material across larger distances. Indeed, DNA becomes limiting in large yeast cells as they are unable to appropriately scale the number of transcripts and proteins to support the increased cellular volume, which leads to inefficient biochemical reactions of the cell and senescence[2]. While precise regulation of myonuclear transcriptional capacity is not understood, there is evidence that mammalian myonuclei exhibit a reserve capacity after muscle overload, and Drosophila myonuclei actively adjust their size and output depending on their location within the myofiber[32,33]. However, what determines flexibility within the myonuclear compartment, and what factors determine a functional transcriptional range have been speculated upon but remain unknown. Currently, there is minimal knowledge of whether DNA is limiting in mammalian muscle cells or under which situations any potential reserve capacity is lost or activated. Overall, the myonuclear domain is a concept premised on early evidence that mRNA transcripts and some myofibrillar protein products are confined to areas close to their orginating myonuclei[34,35]. How such domains are established, the associated flexibility, and a molecular understanding for the roles of new and resident myonuclei in determining these parameters remains to be elucidated. A major reason for this lack of information is that studies investigating the relationship between myonuclear number and skeletal muscle maturity and size, especially in the neonatal period, remain correlational.

Here, we directly manipulate myonuclear accretion through genetic deletion of the muscle fusion factor, Myomaker, at three different neonatal timepoints to generate mice with varied numbers of myonuclei. These novel mouse models allow a more comprehensive analysis of the requirement for multinucleation in mammalian skeletal muscle. Through assessment of the consequences of having fewer myonuclei on establishment of muscle size and function, we uncovered an inherent reserve capacity that can be elicited in myonuclei during development, and is determined by the number of nuclei in the myofiber. Our findings reveal that myonuclear numbers determine not only the ultimate size of the muscle, but also may impact the degree of flexibility it possesses.

## Results

**Development of mouse models with reduced myonuclear number.** We previously showed that deletion of Myomaker (*Mymk*) specifically in adult muscle satellite cells results in fusion incompetence[28,36]. We used these *Mymk*^loxP/loxP; *Pax7*^CreER mice and administered tamoxifen at either P0 or P6 to block fusion in either the first or second week of life (Fig. 1a). Previous work showed down-regulation of *Mymk* transcripts after TAM treatment at P0[37]. Because it is difficult to definitively determine the exact point of fusion-incompetence, we refer to the mice treated at P0 as Δ1w and mice treated at P6 as Δ2w. Analysis of nuclei number in isolated extensor digitorum longus (EDL) myofibers 4 weeks after tamoxifen treatment revealed that Δ1w and Δ2w mice display reduction in myonuclear numbers of 75 or 55%, respectively (Fig. 1b, c). On average, Δ1w myofibers contained 55 myonuclei and Δ2w myofibers contained 100 myonuclei (Fig. 1b, c), which based on previous studies indicates fusion was ablated within the indicated timeframe[20]. The body weights from each group 4 weeks after tamoxifen treatment were reduced compared to control mice (Supplementary Fig. 1a), suggesting some effect on skeletal muscle size. These data show that we successfully titrated the number of nuclei in myofibers through deletion of Myomaker in muscle progenitors during postnatal development.

**Myonuclear numbers impact myofiber size parameters.** The neonatal phase of muscle growth requires myofibers to grow in length and width, with minimal new myofiber formation[20]. To understand the requirement of myonuclear accretion on these parameters we analyzed cross-sectional area (CSA) of myofibers in the tibialis anterior and quadriceps and evaluated myofiber length and volume in isolated myofibers from the EDL. Δ1w mice were analyzed at P28 and Δ2w mice at P35, each 4 weeks after fusion was inhibited, therefore littermate control mice were used at those ages. Histological analysis of both Δ1w and Δ2w

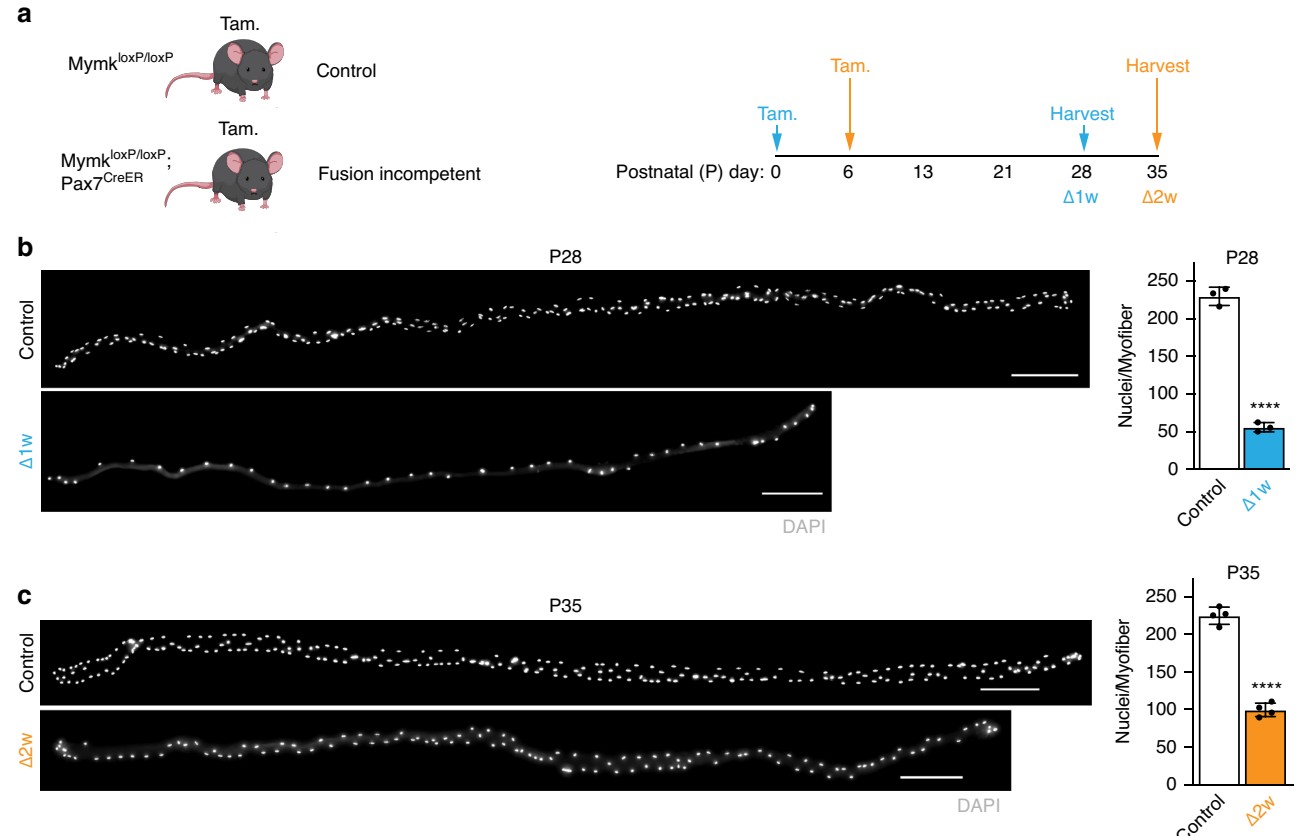

**Fig. 1 Temporal genetic ablation of *Myomaker* in satellite cells during development titrates myonuclear number in myofibers. a** Experimental design used to reduce myonuclear accretion during development. *Mymk*[loxP/loxP] or *Mymk*[loxP/loxP]; *Pax7*[CreER] mice were treated with tamoxifen (Tam.) to generate control or fusion-incompetent mice, respectively. Tamoxifen was administered at either postnatal (P) day 0 (Δ1w) or P6 (Δ2w) and animals were sacrificed 4 weeks post-tamoxifen. **b** Single myofiber images (left) and quantified average number of nuclei per myofiber (right) in control and Δ1w EDL muscle at P28. Control myofibers had an average of 230 nuclei/myofiber, while Δ1w mice had an average of 55 nuclei/myofiber. Myonuclei are labeled with DAPI. **c** Single myofiber images (left) and average number of nuclei per myofiber (right) in Δ2w and control EDL muscle at P35. Control myofibers had an average of 225 nuclei/myofiber, while Δ2w mice had an average of 100 nuclei/myofiber. Myonuclei are labeled using DAPI. In (**b**) and (**c**), $n = 3$ biologically independent animals and 20 myofibers were analyzed per animal. Statistical analyses and data presentation: (**b**) and (**c**), two-sided unpaired *t*-test; ****$P < 0.0001$. Data are reported as mean ± SD. Scale bar: 200 µm. Source data are provided as a Source Data file.

mice revealed normal myofiber architecture, peripherally located nuclei (no central nucleation), and a lack of immune cell infiltration (Fig. 2a, b). Δ1w myofibers were visually smaller compared to control (Fig. 2a). Quantification of CSA in Δ1w mice showed a reduction of 39% in the tibialis anterior (Fig. 2a) and 21% in the quadriceps (Supplementary Fig. 1b). Δ2w mice exhibited a 17% reduction in the tibialis anterior CSA (Fig. 2b) and a non-statistically significant reduction of 15% in the quadriceps (Supplementary Fig. 1c). EDL myofiber length was also impacted by a reduction in myonuclear number, where Δ1w myofibers exhibited a reduction of 21% and Δ2w myofibers were 12% shorter (Fig. 2c, d). We also measured volume of EDL myofibers and observed a 54% reduction in Δ1w and 34% reduction in Δ2w mice (Fig. 2e, f). To understand the kinetics of mouse and muscle growth after genetic blockade of fusion, we analyzed tibia lengths, body weights, and cross-sectional area of myofibers in Δ2w mice at multiple postnatal time points. The tibia lengths and body weights of Δ2w mice exhibited a similar change across time compared to controls (Supplementary Fig. 2a). Tibialis anterior myofiber size remained unchanged between WT and Δ2w mice at P21 and P28, but an increased number of smaller fibers and decreased number of larger fibers were apparent in Δ2w muscle at P42 (Supplementary Fig. 2b), showing that Δ2w mice possess some ability to grow between P28 and P42. These data indicate that a dynamic relationship exists between number of myonuclei and radial

or length size and reveal that muscle can compensate for fewer myonuclei.

**Myonuclear number determines the type of muscle adaptation.** To determine how muscle adapts to fewer myonuclei on a molecular level in the postnatal period we performed microarray analysis on control, Δ1w, and Δ2w muscle at P28. Unbiased clustering from each of the groups using principle component analysis or hierarchical analysis separated samples based on genotype, indicating the three groups of muscle are transcriptionally different and that Δ2w is more similar to control (Fig. 3a, Supplementary Fig. 3). Using gene ontology, we also analyzed differentially expressed genes between control and either Δ1w or Δ2w muscle. While both Δ1w and Δ2w muscle exhibited increased levels of genes associated with muscle development, these differences were more pronounced in Δ1w samples (Fig. 3b). We also observed stark differences in gene expression between Δ1w and Δ2w muscle in terms of fiber type specification and apoptosis/oxidative stress. Δ1w muscle exhibited increased levels of genes present in slow Type I skeletal muscle including *Myh7*, *Tnnc1*, *Atp2a2*, and *Sln* (Fig. 3c). These data indicate that Δ1w muscle does not contain the sufficient number of myonuclei needed for establishment of a normal muscle phenotype.

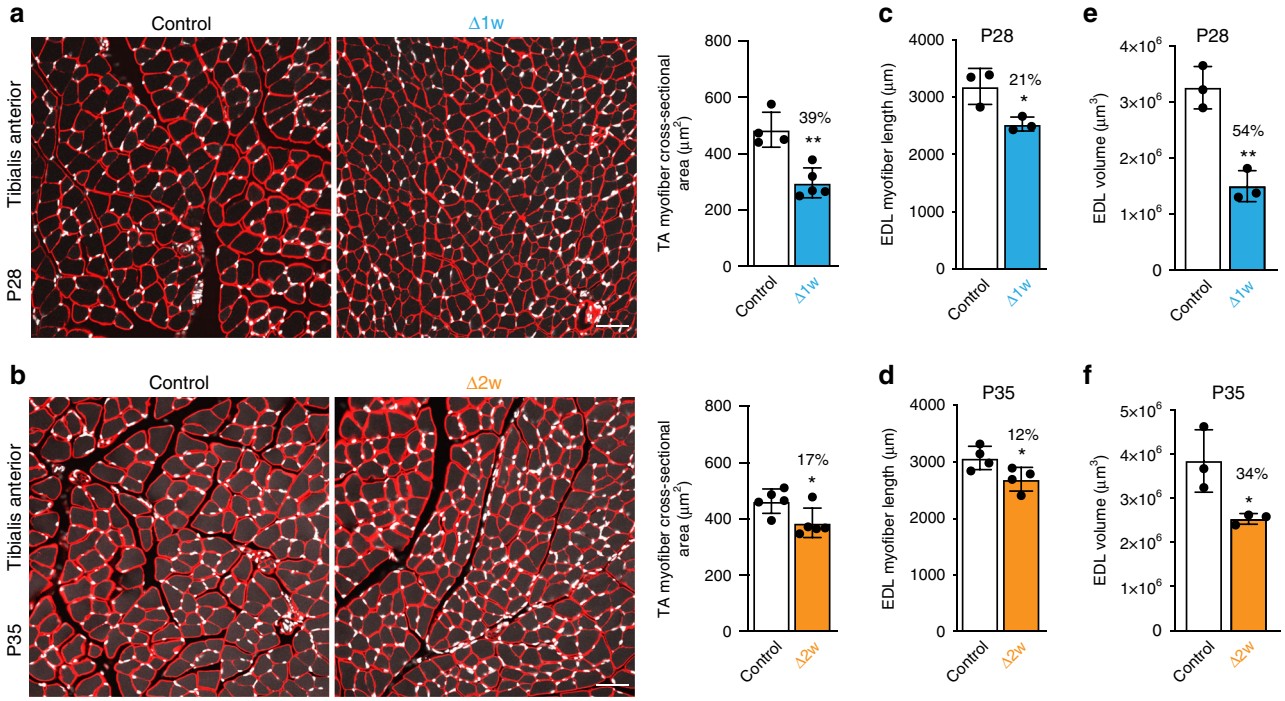

**Fig. 2 Effects of myonuclear reduction on myofiber dimensions. a** Representative sections of tibialis anterior (TA) muscle of Δ1w and control mice at P28 (left). Analysis of cross-sectional area (CSA; right) revealed a 39% reduction in Δ1w myofiber CSA compared to controls. **b** Sections of TA muscle of Δ2w and control mice at P35 (left). CSA analysis (right) showed Δ2w myofibers are 17% smaller than controls. Sections in (**a**) and (**b**) were immunostained with laminin antibodies (red) and also stained with DAPI to label nuclei (white). **c, d** Average length of Δ1w EDL myofibers is decreased by 21% compared to controls (**c**), and Δ2w EDL myofiber length is decreased by 12% (**d**). **e, f** Myofiber volume showed a 54% reduction in Δ1w EDL at P28 (**e**) and a 34% reduction in Δ2w EDL at P35 (**f**). At least 3 20× images were analyzed per mouse in (**a**) and (**b**) ($n = 4–5$ biologically independent animals). 20 myofibers were analyzed per animal in (**c–f**) ($n = 3–4$ biologically independent animals). Statistical analyses and data presentation: (**a–f**) two-sided unpaired $t$-test; *$p < 0.05$, **$p < 0.01$. Data are represented as mean ± SD. Scale bar: 50 μm. Source data are provided as a Source Data file.

Additionally, we observed increased levels of apoptosis and oxidative stress genes in Δ1w muscle suggesting that a 75% reduction of myonuclei results in a maladaptive phenotype (Fig. 3d). Indeed, Δ1w mice die by 200 days of age (Fig. 3e). At 5 months of age, Δ1w mice are hunched and display kyphosis, further indicating that this level of myonuclear number is not sustainable for normal muscle function (Fig. 3f). In contrast, we did not observe obvious kyphosis in Δ2w mice at 5 months of age (Fig. 3f) and no premature death during the duration of this study (Fig. 3e). Overall, the level of myonuclei in Δ1w muscle is not sufficient to establish normal muscle and leads to a maladaptive phenotype, but the level of myonuclei in Δ2w muscle is sufficient to elicit an adaptation that allows survival and relatively normal development under baseline conditions.

**mRNA analysis reveals a range of myonuclear flexibility.** Since Δ1w and Δ2w mice possess fewer myonuclei, but these nuclei lack any genetic perturbation and therefore are genetically normal, we surmised that the adaptive responses elicited in these models could help determine the potential range of functional myonuclear flexibility. To that end, we explored total RNA and mRNA levels of the various groups of mice to understand the relationship between myonuclear number and mRNA output. Total RNA normalized to muscle weight (RNA abundance) revealed no differences between WT, Δ1w, and Δ2w mice at P28 (Fig. 4a). However, when these values were further normalized to the average myonuclear number, a 4-fold increase is revealed in Δ1w muscle and a 2-fold increase in Δ2w muscle (Fig. 4b). These data indicate that in Δ1w and Δ2w muscle there are a greater ratio of RNA:myonuclei compared to WT muscle, suggesting muscles

with fewer nuclei are able to positively regulate RNA abundance. To test if this ability impacted mRNA content, we assessed levels of mRNA transcripts from our microarray analysis and normalized them to total RNA and average myonuclear numbers. The normalization of mRNA transcripts to total RNA yields a metric for mRNA concentration (relative to total RNA content, not volume), which has shown to be size-independent in mononuclear cells[3]. We found that, when compared to controls, Δ1w muscle possessed an increased mRNA concentration of 1.25-fold, whereas Δ2w muscle increased 1.5-fold (Fig. 4c). The greater increase in mRNA concentration in Δ2w muscle is consistent with the idea that they possess a more effective and sustainable adaptive response. Indeed, when assessed on a per nuclear basis (as a proxy for the relative mRNA output per nucleus), this value was appreciably higher in Δ1w muscle (5.3-fold) compared to Δ2w muscle (3.5-fold) (Fig. 4d). We interpret the premature death of Δ1w mice as evidence that this level of transcriptional output is either not sustainable or dysfunctional. In contrast, the absence of premature lethality in Δ2w mice coupled with establishment of smaller but functionally normal muscle (see below) indicates that the increase in relative mRNA output per nucleus in these mice is within a manageable range of flexibility. These data show that myofibers with fewer myonuclei exhibit flexibility in terms of ability to increase mRNA concentrations and also establish a minimal number of nuclei required for muscle development that can sustain life.

**Δ2w mice exhibit larger and functional cytoplasmic domains.** Since Δ1w mice die prematurely, we focused further analysis on a better understanding of the adaptations in Δ2w mice. Specifically,

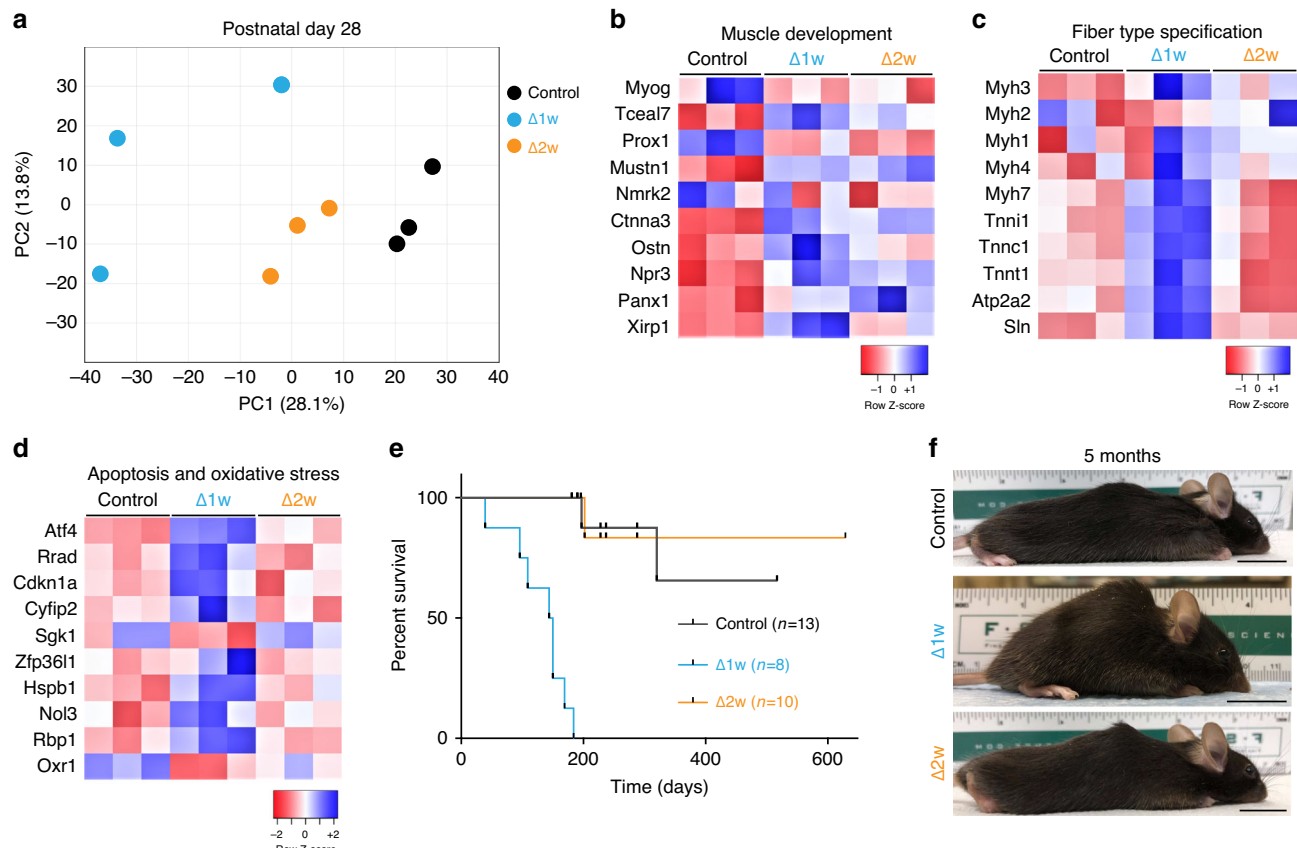

**Fig. 3 Δ1w mice display a maladaptive lethal phenotype in contrast to Δ2w mice. a** PCA plot from microarray analysis showing the differential clustering of Δ1w, Δ2w, and control samples (TA muscle). Δ1w muscle displayed a more divergent profile from that of Δ2w and control muscle, which appeared more similar. **b** Heat map of muscle development genes showed more pronounced alterations in Δ1w compared to Δ2w muscle. **c** Heat map of genes associated with muscle fiber type revealed Δ1w muscle exhibit increased expression of slow fiber type genes (*Myh7, Tnnt1, Atp2a2, Sln*) and reduced expression of fast fiber type genes (*Myh4, Myh1*). **d** Heat map showing increased expression of apoptosis and oxidative stress genes in Δ1w muscle. **e** Survival curve for Δ1w, Δ2w, and control mice. Δ1w mice began dying prematurely prior to three months of age while Δ2w mice did not. **f** Images of Δ1w, Δ2w, and control mice showed kyphosis in Δ1w mice, while Δ2w mice looked similar to controls. Data presentation: (**a–d**) data generated using microarray data performed on TA muscle at P28; *n* = 3 per group. Scale bar: 2 cm. Source data are provided as a Source Data file.

we asked: (1) if the compensation observed during early development is sustained and able to support maturational growth (from adolescence to adulthood), when there is limited addition of new myonuclei, and (2) how the myonuclear domain is impacted by the the reduction in myonuclei numbers. We began by analyzing isolated EDL myofibers from control and Δ2w muscle at multiple time points from P13 to P150. This approach allowed determination of nuclear number, cross-sectional area, and myonuclear domain from the same sample material. A 200–400-μm segment from at least 30 myofibers per animal was imaged and rendered in 3D (Fig. 5a), which was then utilized for quantification of multiple parameters. The myonuclear number in Δ2w muscle was reduced at P14, one week after ablation of fusion, and remained reduced through P150 (Fig. 5b). These data show that the myonuclear number in Δ2w muscle is not rescued through karyokinesis of existing myonuclei or through fusion of a Pax7-independent stem cell population. The cross-sectional area of EDL myofibers begins to diverge at P35 in Δ2w muscle and remains reduced at P42 and P150 (Fig. 5c). However, between P42 and P150 we observed an increase in CSA of both control and Δ2w muscle suggesting that muscle with fewer nuclei can grow in width during the maturational phase of muscle growth. Analysis of myonuclear domains in the isolated EDLs revealed larger cytoplasm:DNA ratios in Δ2w muscle beginning at P28, and this difference increased at P150 (Fig. 5d). These data

indicate that muscle fibers with a 55% reduction in nuclei and expanded cytoplasm:DNA ratios can grow radially during the maturational growth phase of muscle development.

As an independent method to determine the ability of Δ2w mice to grow and function during maturation from P42 to adulthood, we analyzed cross sections of the tibialis anterior and evaluated function of hindlimb muscles. Consistent with our findings in the EDL, tibialis anterior cross-sectional area of Δ2w muscle increased by a similar percentage as control between P42 and P150 (Fig. 5e). Assessment of muscle function using an in situ approach that directly measures force produced by the tibialis anterior revealed reduced peak force in Δ2w mice, consistent with their muscles being smaller than controls (Fig. 5f). However, specific force (peak force normalized to physiological cross-sectional area) and fatigue were comparable between control and Δ2w mice (Fig. 5f). Moreover, functional analysis on skinned single myofibers from the TA revealed similar specific force between control and Δ2w mice (Fig. 5g). One aspect of Δ2w muscle that could compensate for function are changes in fiber type, but we observed normal distributions of myosin isoforms (Supplementary Fig. 4).

Thus, our combined functional analysis indicates normal myofiber quality in Δ2w mice. Consistent with normal specific force, Δ2w muscles contain normal relative levels of contractile proteins that comprise the muscle sarcomere (Fig. 5h). Taken together, Δ2w muscle harbors a 55% reduction of myonuclei

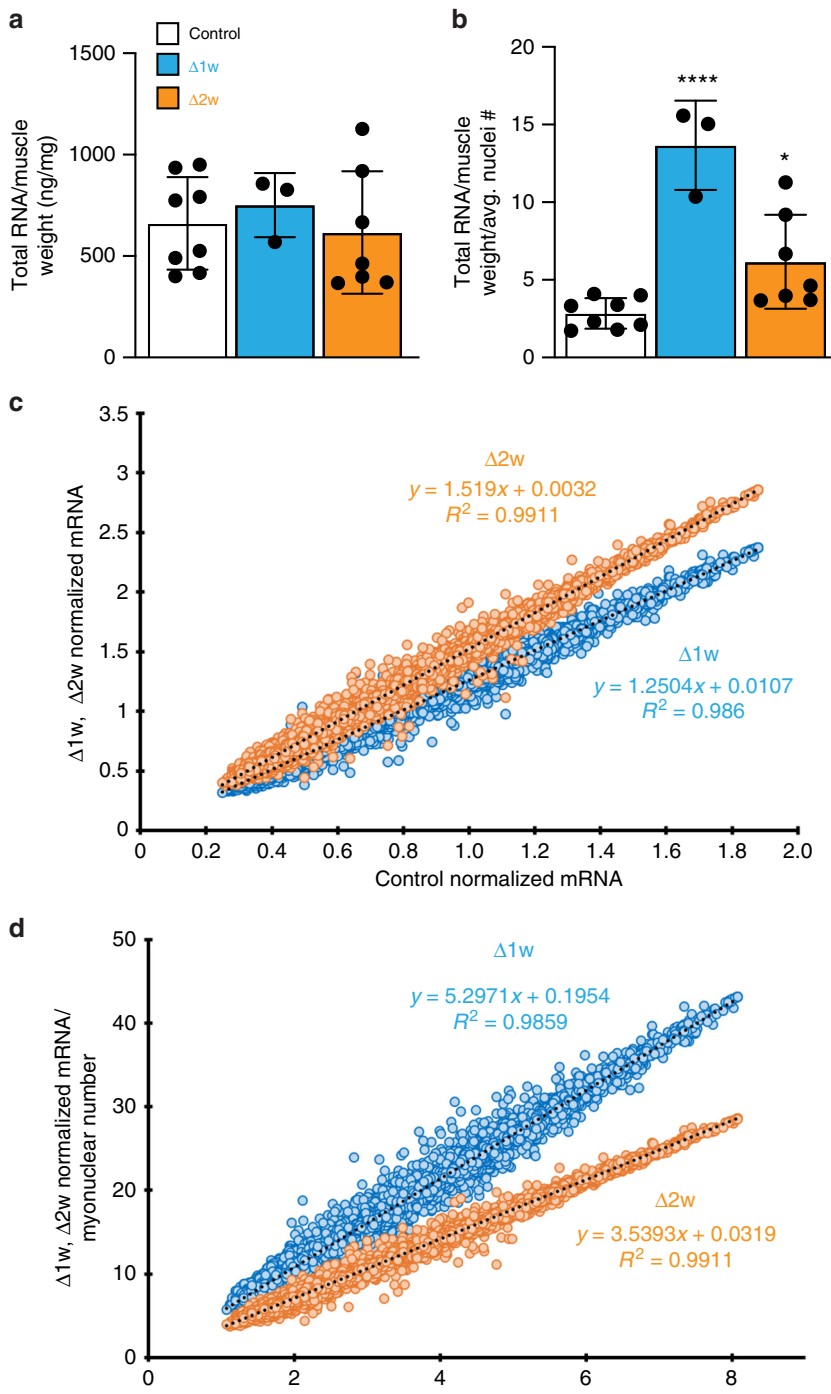

**Fig. 4 Myofibers adapt to the reduction in myonuclear numbers by elevating mRNA concentrations. a** Total RNA levels from the tibialis anterior normalized to muscle weight for the indicated groups of mice at P28. **b** Normalized RNA levels (from **a**) further normalized to the average number of nuclei/myofiber. For (**a**) and (**b**), $n = 3$–8 biologically independent animals. **c** Comparison of the mRNA concentration (normalization of sst-rma signals from microarray of 22,207 annotated transcripts to total RNA for the respective sample) of Δ1w and Δ2w muscle to control revealed a greater contribution of mRNA to total RNA (mRNA concentration) being generated in both Δ1w (slope = 1.2504) and Δ2w (slope = 1.519) muscle (represented as the slope of the best-fit line). **d** Data from **c** were normalized further to average number of nuclei/myofiber, revealing an even larger increase in the percentage of mRNA:total RNA being generated on a per nucleus basis in Δ1w (slope = 5.2971) and Δ2w (slope = 3.5393) muscle compared to controls. Statistical analyses and data presentation: (**a**) and (**b**), one-way ANOVA with a Tukey correction for multiple comparisons; *$p < 0.05$, ****$p < 0.0001$. Significance compared to control group. Data are reported as mean ± SD. Source data are provided as a Source Data file.

leading to smaller myofibers, but expanded myonuclear domains and the ability to develop specific force comparable to controls. These data establish that a reserve capacity to support expanded myonuclear domains and functional growth exists in myonuclei during postnatal development.

**Myonuclear number limits myofiber flexibility**. We have shown that Δ2w mice consistently possess an increased mRNA concentration (relative to total RNA content, not volume) in muscle, even though they contain fewer nuclei. Thus, compared to control mice, Δ2w mice are able to maintain higher transcript

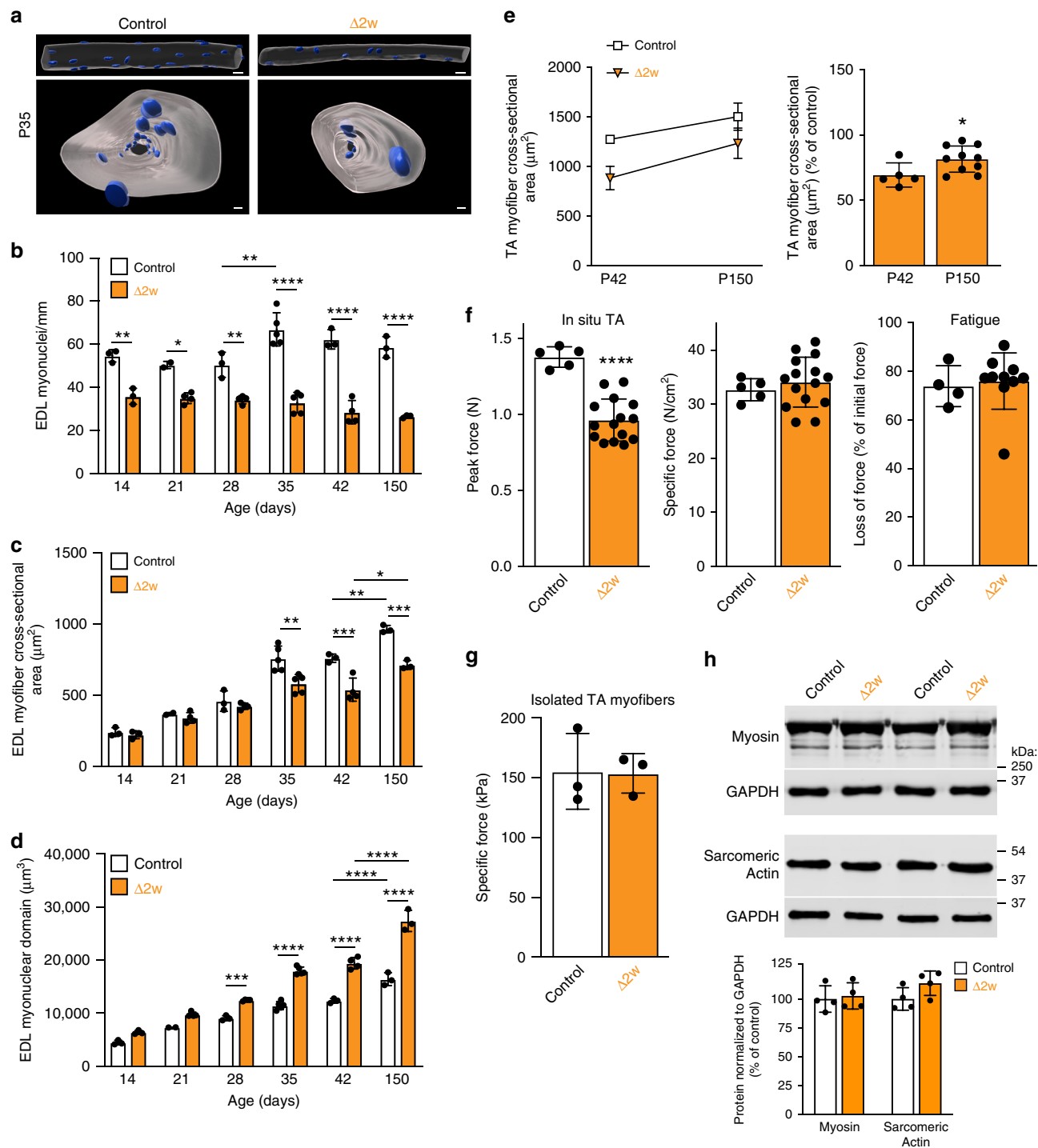

concentrations and larger myonuclear domains. One potential explanation of these data is simply that a reduction in myonuclear numbers elicits the adaptive flexibility observed in the Δ2w mice. To further test the relationship between myonuclear number and flexibility to establish size, we generated a mouse model with more nuclei than Δ2w mice but less than control. To achieve this, we treated $Mymk^{loxP/loxP}$; $Pax7^{CreER}$ or $Mymk^{loxP/loxP}$ (as controls) with tamoxifen at P13 (Fig. 6a), the final week during development where robust myonuclear addition takes place. In these mice, which we refer to as Δ3w, analysis of myonuclear number in isolated EDLs showed that these myofibers exhibit a 25% reduction of nuclei (Fig. 6b). Thus, Δ3w muscle have a nuclear number (approximately 180/myofiber) between control

(233/myofiber) and Δ2w (100/myofiber). Body weights of Δ3w mice were smaller compared to controls (Supplementary Fig. 5a). We analyzed size parameters of muscle in Δ3w mice at P42, 4 weeks after ablation of fusion, and found a 22–24% reduction of cross-sectional area in the tibialis anterior and quadriceps (Fig. 6c, Supplementary Fig. 5b). We also observed a 9% reduction of EDL length in Δ3w mice (Fig. 6d) and a 33% reduction in volume (Fig. 6e).

To directly compare size regulation in Δ2w and Δ3w muscle we analyzed cross-sectional area at P42 as a percentage of littermate controls. Here we found that both Δ2w and Δ3w muscle exhibited a reduction in size despite Δ3w muscle containing more myonuclei (Fig. 7a). Moreover, the myonuclear domain in

**Fig. 5 Δ2w mice retain the ability to generate normal muscle quality with enlarged myonuclear domains. a** Representative images showing 3D rendered myonuclei (blue) and myofiber volume (gray) from a section of control and Δ2w EDL myofibers at P35, viewed from the side (top) and through (bottom) the myofiber. Images were used to quantify nuclei number, myofiber area and myonuclear domain. **b** Number of myonuclei/mm in EDL myofibers from the indicated groups of mice at various ages. **c** EDL myofiber cross-sectional areas in control and Δ2w muscle. **d** Quantification of changes in myonuclear domain in control and Δ2w EDL myofibers over time. **b**–**d** $n = 3$–5 biologically independent animals. **e** Comparison of tibialis anterior (TA) cross-sectional area at P42 and 5 months, expressed as growth over time (left), and percentage of control (right), revealed the retained ability of Δ2w muscle to grow at the same rate as control muscle. $n = 5$–10 biologically independent animals. **f** Control and Δ2w muscle were subjected to in situ muscle force measurements to assess functional capacity at five months of age. Δ2w tibialis anterior generated a lower peak tetanic force (left), but when normalized to physiological cross-sectional area, Δ2w muscle exhibited similar specific force as controls (middle), indicating the overall functional quality of muscle in the two groups is the same. Fatigue, measured as loss of initial force following continuous isometric stimulations, was similar between the two groups (right). $n = 5$–15 biologically independent animals. **g** Contractile function of single myofibers from control and Δ2w tibialis anterior muscles. 6 myofibers were analyzed per animal. $n = 3$ biologically independent animals. **h** Representative western blots for myosin and sarcomeric actin from control and Δ2w quadriceps (top). Quantification of the protein signal normalized to GAPDH (bottom). $n = 4$ biologically independent animals. Statistical analyses and data presentation: (**b**–**d**) two-way ANOVA with a Tukey correction for multiple comparisons. **e**–**h** Two-sided unpaired $t$-test; *$p < 0.05$, **$p < 0.01$, ***$p < 0.001$, ****$p < 0.0001$. Data are reported as mean ± SD. Scale bars: (**a**) 30 μm (top), 10 μm (bottom). Source data are provided as a Source Data file.

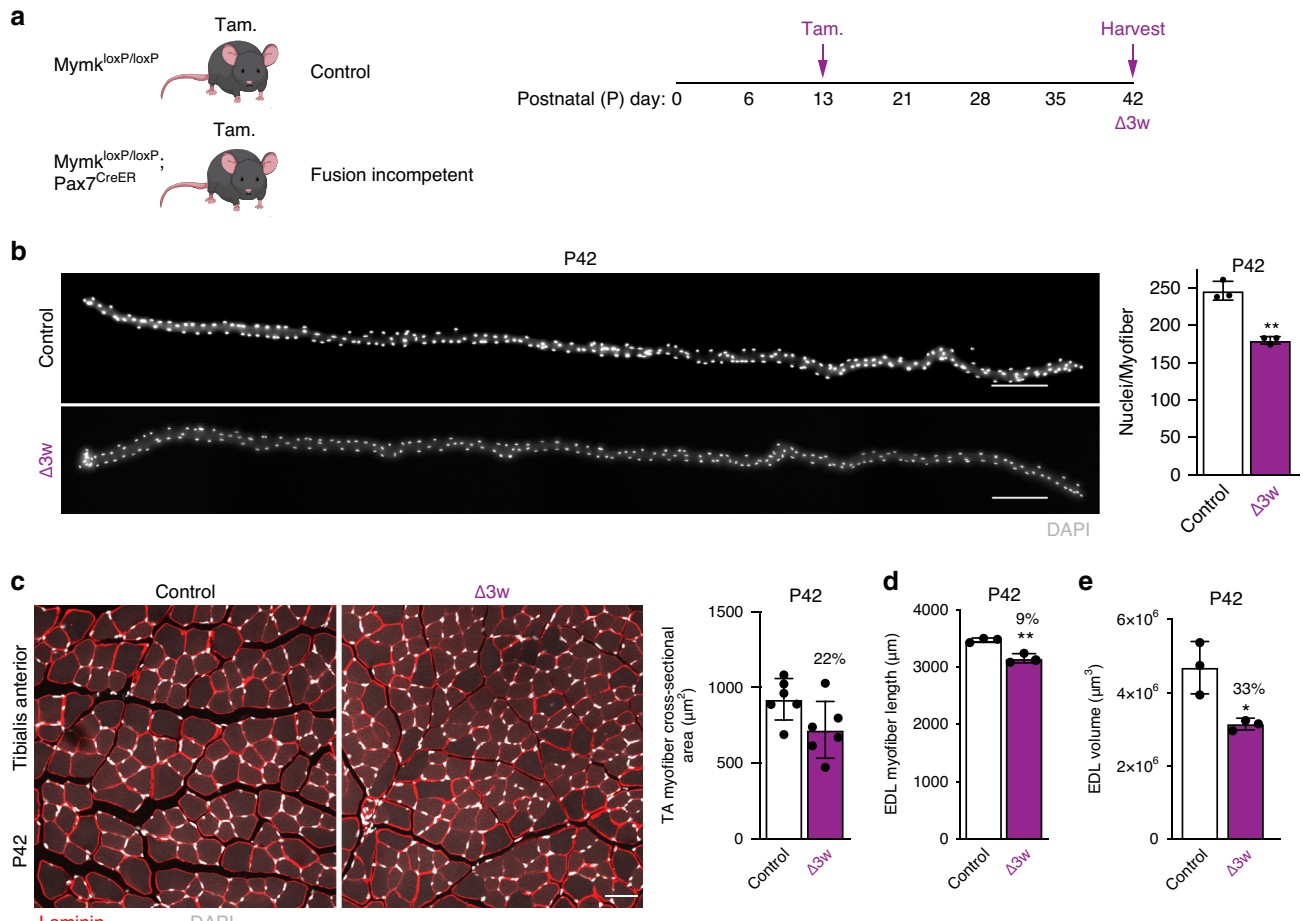

**Fig. 6 Size characteristics in Δ3w mice. a** Experimental design used to generate a third genetic model, Δ3w mice. The indicated groups of mice were administered tamoxifen (Tam.) at P13 and tissue was collected four weeks later. **b** Single myofiber images (left) and quantified average number of nuclei per myofiber (right) in Δ3w and control EDL muscle at P42. Control myofibers have an average of 246 nuclei/myofiber, while Δ3w mice have an average of 180 nuclei/myofiber. Myonuclei are labeled with DAPI. **c** Representative sections of Δ3w and control tibialis anterior (TA) muscle at P42 (left). Analysis of cross-sectional area (right) showed Δ3w myofibers are 22% smaller than controls. Sections in (**c**) were immunostained with laminin antibodies (red) and stained with DAPI (white). **d** Average length of Δ3w EDL myofibers is decreased by 9% compared to controls. **e** Myofiber volume showed a 33% reduction in Δ3w EDL at P42. 20 myofibers were analyzed per animal ($n = 3$ biologically independent animals) in (**b**), (**d**), and (**e**). At least 3 20× images were analyzed per mouse in (**c**) ($n = 6$ biologically independent animals). Statistical analyses and data presentation: (**b**–**e**) two-sided unpaired $t$-test; *$p < 0.05$, **$p < 0.01$. Data are represented as mean ± SD. Scale bars: (**b**) 200 μm, (**c**) 50 μm. Source data are provided as a Source Data file.

isolated EDL myofibers was not different from control myofibers (Fig. 7b). Δ3w mice possess the same myonuclear domain as controls, but with fewer nuclei, resulting in smaller muscle. That Δ2w and Δ3w muscle exhibited similar size, but with divergent effects on cytoplasm:DNA ratios, prompted us to compare mRNA concentrations per nucleus, as a measurement of flexibility, between the two models. At P42, there were no differences in total RNA of the tibialis anterior when normalized

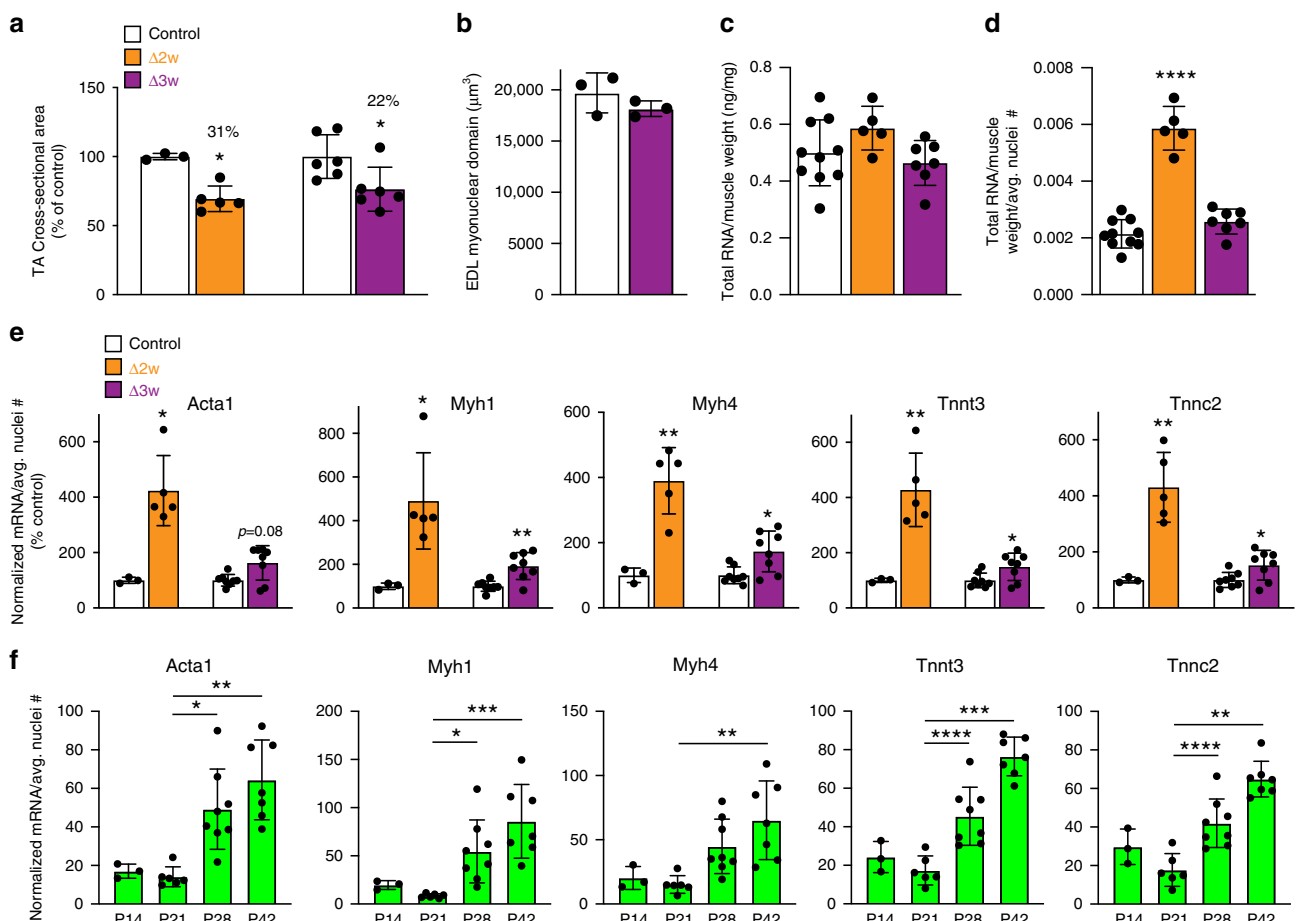

**Fig. 7 Flexiblity is inversely correlated with myonuclear number. a** A comparison of the cross-sectional area of tibialis anterior muscle in Δ2w and Δ3w mice at postnatal (P) day 42 revealed similar effects on cell size, despite differences in myonuclear number ($n = 3–7$ biologically independent animals). **b** Assessment of the myonuclear domain in isolated EDL myofibers from Δ3w mice at P42 ($n = 3$ biologically independent animals). **c** Total RNA levels normalized to muscle weight in the various groups of mice ($n = 5–10$ biologically independent animals). **d** Normalized RNA levels from (**c**) were normalized again to the average number of nuclei/myofiber ($n = 5–10$ biologically independent animals). A significant increase in the amount of RNA per myonucleus is observed in Δ2w muscle, but remains unchanged in Δ3w muscle. **e** Concentration of transcripts coding for key skeletal muscle structural genes (*Acta1*, *Myh1*, *Myh4*, *Tnnt3*, and *Tnnc2*) on a per nuclear basis are increased in Δ2w and Δ3w muscle compared to controls ($n = 3–8$ biologically independent animals). Δ2w muscle has a larger increase in transcript concentration per nucleus than Δ3w, despite having fewer myonuclei per fiber. To obtain these values, relative transcript levels were determined by semi-quantitative qPCR from P42 samples, which were then normalized to total RNA and average myonuclear numbers for each genotype. **f** Abundance of transcripts on a per nuclear basis in wild-type tibialis anterior across developmental time points ($n = 3–7$ biologically independent animals). Statistical analyses and data presentation: **a**, two-sided unpaired *t*-test. **c**, **d** One-way ANOVA with a Tukey correction for multiple comparisons; significance compared to control group. **e**, *Acta1* (both groups), *Myh1* (Δ2w only): Mann–Whitney test. *Myh1* (Δ3w only), *Myh4*, *Tnnt3*, and *Tnnc2* (both groups): two-sided unpaired *t*-test with Welch's correction. **f** One-way ANOVA with Tukey correction for multiple comparisons. *Acta1* was analyzed with one-way ANOVA with Kruskal–Wallis correction for multiple comparisons. *$p < 0.05$, **$p < 0.01$, ***$p < 0.001$, ****$p < 0.0001$. Data are represented as mean ± SD. Source data are provided as a Source Data file.

to tissue weight between control, Δ2w, and Δ3w mice (Fig. 7c). When this value was further normalized to average myonuclear number, we observed an increase in Δ2w muscle, confirming our observations obtained at P28; in contrast, no such increase was detected in Δ3w muscle (Fig. 7d). While relative output of sarcomeric transcripts on a per nuclear basis were increased in both Δ2w and Δ3w muscle, the magnitude was modest in Δ3w compared to Δ2w muscle (Fig. 7e). These data reveal that a reduction of myonuclear numbers elicits an increase in mRNA output on a per nuclear basis in both Δ2w and Δ3w mice. However, Δ3w muscles, which contain 1.8-fold more nuclei, do not achieve the magnitude of output observed in Δ2w muscles. These data indicate that Δ3w muscle also has an ability to increase relative mRNA output compared to controls, but not to levels sufficient to enlarge myonuclear domains or establish muscle size to control levels. Overall, we interpret the diminished

adaptive response observed in Δ3w muscles to indicate that myonuclear flexibility (an ability to sustainably enhance mRNA concentrations and support larger cytoplasmic volumes), seems to be determined directly by the number of myonuclei.

Cellular volume has also been proposed to be sensed by nuclei and result in increased output[38]. While our data indicate that nuclei number itself regulates output, we sought to also test the effect of volume, in a setting where nuclei numbers are not increased, on myonuclear flexibility. Genetic and pharmacologic inhibition of myostatin, a negative regulator of muscle mass, leads to increases in myofiber size but does not elicit myonuclear accretion[24,29,39,40]. We treated 3-month-old control and Δ2w mice with ACVR2B-Fc, a decoy receptor for myostatin, over 4 weeks (Supplementary Fig. 6a). Control and Δ2w mice exhibited hypertrophy of myofibers to similar levels (Supplementary Fig. 6b, c). No increase in mRNA output per nucleus

(normalizing mRNA to average numbers of myonuclei) for sarcomeric genes was observed after ACVR2B-Fc treatment in either control or Δ2w mice (Supplementary Fig. 6d). The failure to increase mRNA output by ACVR2B-Fc may explain the reduced muscle quality in animals where myostatin has been inhibited[24,40]. These data suggest that stimuli which lead to increases in myofiber volume, such as with myostatin inhibition, cannot be assumed to drive global increases in mRNA output. Overall, our data indicate that the number of nuclei predominate, over other factors such as volume, in determining myonuclear flexibility and mRNA concentrations.

The data presented thus far indicate that addition of myonuclei do not always yield an increase in output per nucleus, but these observations were made in genetic models. When extrapolated to normal postnatal muscle development, the sum total of our data suggest a paradigm where increase in myonuclear numbers during this period does not correlate with increase in cellular total mRNA abundance. If true, increases in mRNA abundance on a per nuclear basis should be limited during these postnatal developmental phases. To test this paradigm, we analyzed transcript abundance of *Acta1*, *Myh1*, *Myh4*, *Tnnt3*, and *Tnnc2* in control muscle during developmental myonuclear accretion (P14 and P21) and also, after the majority of that process had terminated (P28 and P42). Note that we were not assessing myonuclear flexibility here because there is no genetic perturbation to reduce nuclei numbers, and therefore did not measure mRNA concentrations. Consistent with our hypothesis, we found that despite an increase in myonuclear numbers between P14 and P21, transcript abundance on a per nuclear basis did not increase for any of the genes analyzed (Fig. 7f). Increases detected at P28 and P42 (Fig. 7f) indicate that myofibers retain the ability to globally upregulate mRNA abundance from resident myonuclei during maturational growth.

## Discussion
Our study begins to unravel the relationship between myonuclear number, achievement of target size, and determinants of myofiber flexibility in multinucleated fast skeletal muscle. The results presented here show that myonuclear numbers are the ultimate determinant of size in mammalian skeletal muscles. However, our data indicate that just increasing the number of myonuclei would not result in a commensurate increase in muscle size or volume. Therefore, the relationship between myonuclei numbers and myofiber size is not a simple linear one, a concept that has been independently validated in an accompanying manuscript[41]. Through analysis of mRNA concentration and output per nucleus in mice where myonuclei numbers were titrated, we determined a range of myofiber flexibility that could sustain muscle quality. On a per nucleus basis, Δ1w mice generated more mRNA but had a maladaptive compensation with reduced mRNA concentration compared to Δ2w mice. These data suggest that dysregulation of the mRNA transcriptional profile in Δ1w mice may underlie the kyphosis and early lethality and indicate that these mice have not reached the threshold of myonuclear numbers necessary for establishing functional muscle. In contrast, Δ2w mice are able to grow and maintain functional muscle, indicating that the increase in mRNA concentration and output per nucleus is within a healthy and sustainable range. These findings suggest a threshold number of myonuclei that are necessary for their reserve capacity to be effectively utilized to support myofiber flexibility during development.

Despite exhibiting smaller myofibers with reduced absolute force, Δ2w mice display normal gross structure and development of specific force comparable to controls. In an accompanying manuscript[41], we show that the larger fibers of the Δ2w mice display less DNA content (nuclear number) per fiber volume

relative to smaller fibers, and thus exhibit a type of scaling that takes the form of power laws. Specifically, nuclear number maintains a one-to-one pace with fiber surface area, but nuclei are able to sustain higher functional fiber volumes with increments in myofiber size. However, we are unable to exclude that other aspects of muscle function could be impaired by the reduced number of nuclei. For instance, these mice might be more impacted during aging or they may not be able to respond appropriately to stresses such as exercise. It is difficult to assess if the Δ2w mice could respond to exercise since that stimulus may acutely require satellite cell fusion[26,28,29], and the Δ2w mice not only possess fewer myonuclei but also an inability to add new myonuclei in the adult because the deletion of Myomaker in satellite cells is indelible. New mouse models and techniques will be required to fully explore the limits of myofiber flexibility in mice where myonuclear numbers are titrated.

One of the most interesting aspects of our study is the comparison between Δ2w and Δ3w mice. We anticipated that the deficiency of myonuclear numbers in Δ3w mice would elicit an adaptive response as was observed in Δ2w mice, which have a 55% reduction in myonuclei but achieve 65% of WT volume, with an associated increase in mRNA concentrations. We reasoned that such an adaptive upregulation in transcriptional output in a larger population of myonuclei (Δ3w mice had a 1.8-fold increase in myonuclei compared to Δ2w mice) would result in full compensation and establishment of WT myofiber size. However, while evidence of elevated mRNA concentrations was found in Δ3w mice, those increases do not remotely reach what was achieved in Δ2w mice, and Δ3w mice do not achieve a similar size as controls. Thus, the inability to establish normal WT size in Δ3w mice is a consequence of having fewer nuclei and inadequate flexibility. An alternative explanation is that the identity of nuclei, not overall number, is the primary factor driving the differences in flexibility between Δ2w mice and Δ3w mice. However, since Δ3w mice exhibit some flexibility, just less than what is achieved in Δ2w mice, we propose that myonuclear number controls adaptation through modulation of overall flexibility. Specifically, that when more nuclei are added to the syncytium, overall output may increase, but each nucleus contributes progressively less.

This paradigm suggests a potential cost of having myofibers rich in nuclei is that it limits the range of myonuclear flexibility, indicating an unexpected inverse relationship between myonuclear numbers and an ability to upregulate mRNA concentration. This could be caused by direct negative influences between the nuclei or may result from increasing nuclei numbers competing for essential but limited transcription factors. Work on mammalian cells forced to form heterokaryons has implicated the existence of a factor that helps modulate transcription, providing support for the idea that nuclei can negatively regulate output from other nuclei in a syncytium[38]. The identity of a factor that could regulate transcriptional output between nuclei is not known in any system. Another question that remains to be answered is whether the adaption of increased mRNA concentration in Δ2w mice is due to enhanced mRNA production and/or diminished mRNA degradation. Also, elegant work in Drosophila showed that myonuclei exhibit the ability to endoreplicate[33], but endoreplication has not been detected during normal postnatal development in mammals rats[42,43]. Consistent with mammalian myonuclei not being able to endoreplicate, we did not detect changes in nuclear volume, an indicator of genome content, in Δ2w mice[41].

Our models where myonuclear numbers have been titrated may explain the conundrum of myonuclear flexibility during development and inflexibility in the adult. Based on multiple studies where the function of muscle stem cells was genetically inhibited, accrual of new myonuclei is required for adult

adaptations including hypertrophy[26–31]. Thus, in the adult there seems to be limited flexibility of hundreds of myonuclei that were added during development to increase mRNA concentration, although there is an ability to upregulate rRNA[44]. The divergence in outcomes between the Δ2w and Δ3w models reveal that having more nuclei limits overall myofiber flexibility. The retention of such limiting mechanisms in adult myofibers would account for the observed reliance on myonuclear accretion for adaptability, and suggest that acquisition of new nuclei to adapt is more economical than globally upregulating transcription in all resident myonuclei. Specifically, in situations where the stimulus is mild or localized to a specific domain of the myofiber, such as exercise-induced injury[29], developmental growth in length[20], and regulation of the neuromuscular junction[45], accrual of nuclei from the stem cell population may preferentially occur as global upregulation of transcriptional output is unnecessary and wasteful. In contrast, in situations where the stimulus is sensed throughout the myofiber, such as maturational growth (P21 to adult), where global upregulation of mRNA output is necessary, hypertrophy occurs without an absolute reliance on accrual of new nuclei. While myofibers clearly have the potential to establish domains of myonuclei that harbor different transcriptional profiles, such as at the neuromuscular junction and myotendinous junction[12,46–48], myofibers may not possess the ability to selectively modulate transcriptional output from resident myonuclei in microdomains related to properties that are more globally regulated such as radial size and force.

In this report, we study growth of muscle in multiple dimensions. Growth of muscle length during neonatal development coincides with the timing of myonuclear accretion, therefore we were surprised by the stronger effect of myonuclear titration on radial size compared to length in hindlimb muscles. We previously showed that Δ1w mice do not develop neuromuscular contractures, a syndrome associated with shorter skeletal muscle[37]. However, shorter myofibers could be compensated for by pennation angles and tendon length, yielding a functional joint absent of contractures. Overall, the relatively mild effect on myofiber length with only 25% of WT myonuclei could be explained by the idea that there is maximal pressure on the muscle to grow in length during this period because of bone growth. Thus, while normal length of individual myofibers require the full amount of nuclei this can be compensated for to make sure muscle growth keeps pace with bone growth.

One limitation of our study is that we mainly focus on the regulation of mRNA for muscle growth and do not deeply investigate protein synthesis and degradation, which are critical for regulation of muscle size[49,50]. The reason for the focus here on mRNA is that the main perturbation in our models is at the level of nuclei, which determine the number of templates available for protein generation. Relative myosin and sarcomeric actin levels were unchanged in Δ2w muscle when normalized to GAPDH. However, when these protein levels were normalized to muscle weights, Δ2w muscle would exhibited an increase (~1.5-fold; not shown) that mirrored the increase observed in mRNA concentrations. This could imply a proportional increase in both transcript and protein levels of key contractile factors, which exceed the amount of protein incorporated into functional sarcomeres. Overall, we anticipate that future work will reveal proteostasis-related adaptations in the Δ2w mice.

Finally, these results also advance our knowledge of myonuclear domains in myofibers, the defined cytoplasmic volumes controlled by each of the hundreds of resident myonuclei. We propose that during postnatal development, robust myonuclear accretion correlates with myofiber growth in length. Once the majority of length growth has been achieved, there is not a requirement for robust myonuclear accretion, but now muscle has a plethora of nuclei that can contribute to maturational growth. Our results indicate that the magnitude of that contribution is defined by the mRNA output on a per nuclear basis, the range of which appears to be determined by the number of myonuclei present. The limits of this range define the size of myonuclear domains that can be supported during maturational growth, or regrowth in adults after hindlimb suspension is reversed, where there is significant hypertrophy without accretion of new myonuclei[51,52]. The paradigm that nuclei within each fiber limits flexibility of individual resident myonuclei would account for the near obligatory reliance of adult muscle myofibers on myonuclear accretion for adaptations. Future experiments targeting transcriptional output will clarify this and other concepts highlighted by the current work, with the potential to redefine our understanding of myonuclear domains, size regulation, and adaptive flexibility in multinucleated muscle cells. Understanding the molecular circuitry for the reserve capacity in our models with titrated myonuclear numbers could help design novel strategies to maintain muscle mass without the need for accretion from the muscle stem cell pool.

## Methods

**Animals and tamoxifen treatment**. To ablate myomaker specifically in muscle satellite cells, $Myomaker^{loxP/loxP}$ mice were bred with mice carrying the muscle stem cell-specific $Pax7^{CreER}$ conditional Cre recombinase. $Myomaker^{loxP/loxP}$; $Pax7^{CreER}$ mice represented the experimental group, while $Myomaker^{loxP/loxP}$ mice served as controls. Tamoxifen (Sigma-Aldrich) was dissolved in corn oil with 10% EtOH and administered to all mice by intraperitoneal (IP) injections. Neonatal dosage of tamoxifen was 200 μg (Δ1w and Δ2w mice) or 300 μg (Δ3w mice). For all groups, equal ratios of both male and female mice were used. Mice were housed in a room with an ambient temperature of 72 °F, with 30–75% humidity, and a 10/14 h dark/light cycle. All animal procedures were approved by Cincinnati Children's Hospital Medical Center's Institutional Animal Care and Use Committee.

**Muscle collection and preparation**. To collect muscles for analysis, mice were anesthetized using isoflurane and then cervical dislocation was performed. Muscles were harvested, weighed, and then either flash frozen in liquid nitrogen for molecular analysis, or embedded in tragacanth and frozen in liquid nitrogen-chilled 2-methylbutane for downstream histological analysis. Some cohorts of muscle (both control and experimental) were collected and fixed in 4% paraformaldehyde (PFA)/PBS for 3–4 h, then cryoprotected in 30% sucrose/PBS overnight. Muscles were then embedded in OCT and frozen in liquid nitrogen-chilled 2-methylbutane prior to sectioning. All muscles used for histology were cryosectioned at 10 μm thickness at the muscle mid-belly.

**Immunohistochemistry and quantification of myofiber cross-sectional area (CSA)**. Muscle sections were first fixed in 1% PFA for 2 min, washed in PBS, permeabilized in 0.2% Triton X-100/PBS for 10 min, washed in PBS, and blocked in 1% BSA/1% heat inactivated goat serum/0.025% Tween20/PBS for 1 h at room temperature in a humidity chamber. Next, slides were incubated in rabbit anti-laminin primary antibody (1:100; Sigma-Aldrich) for 1 h at room temperature, or overnight at 4 °C, in a humidity chamber. After incubation in primary antibody, slides were washed in PBS, and incubated in goat anti-rabbit Alexa Fluor 647 secondary antibody (1:200; Invitrogen) for 30–60 min at room temperature in a humidity chamber. Slides were then washed with PBS and mounted with VectaShield containing DAPI (Vector Laboratories). After immunostaining, representative images were taken on a Nikon A1R confocal microscope for use in CSA analysis. The CSA of individual myofibers were measured using NIS Elements software (Nikon), which can quantify the area within laminin-labeled myofibers. After all CSA measurements were collected, CSA distribution graphs were generated by binning all data to produce a parametric distribution in control groups, and then comparing experimental distributions using the same binning.

**Immunohistochemistry and quantification of fiber type distribution**. Muscle sections were first fixed in 1% PFA for 2 min, washed in PBS, permeabilized in 0.2% Triton X-100/PBS for 10 min, washed in PBS, and blocked in 1% BSA/1% heat inactivated goat serum/0.025% Tween20/PBS with AffiniPure Fab Fragment goat anti-mouse IgG diluted 1:50 for 2 h at room temperature in a humidity chamber. Next, slides were co-incubated with primary antibodies against MYH7 (1:50; BA-D5, DSHB), MYH2 (1:20; SC-71, DSHB), MYH4 (1:10; BF-F3, DSHB), and rabbit anti-laminin (1:100; Sigma-Aldrich) overnight at 4 °C in a humidity chamber. After incubation in primary antibodies, slides were washed in PBS, and co-incubated in goat anti-mouse IgG2b Alexa Fluor 594 (1:100; Invitrogen), goat anti-mouse IgG1 Alexa Fluor 647 (1:100; Invitrogen), goat anti-mouse IgM Alexa

Fluor 488 (1:100; Invitrogen), and goat anti-rabbit Alexa Fluor 405 (1:200; Invitrogen) secondary antibodies for 90 min at room temperature in a humidity chamber. Slides were then washed with PBS and mounted with VectaMount permanent mounting medium (Vector Laboratories). After immunostaining, representative images were taken on a Nikon A1R confocal microscope for use in fiber type analysis. Fibers were analyzed for which antibody was stained most dominantly, with unlabeled myofibers counted as TypeIIx. Analysis was completed using FIJI (ImageJ).

**Single myofiber isolation and nuclei count**. EDL muscles were collected and incubated in high-glucose DMEM (HyClone Laboratories) with 0.2% collagenase type I (Sigma-Aldrich) at 37 °C in a cell culture incubator. After 30–45 min of incubation, muscles were gently triturated using a wide bore glass pipette to loosen digested myofibers, and then returned to the incubator. After no more than a 1-hour incubation, muscles were triturated (first using a wide bore, then a small bore pipette) until myofibers shed from muscle. Single myofibers were collected and fixed in 4% PFA/PBS for 20–30 min at room temperature, and subsequently stored in PBS at 4 °C. To immunostain nuclei with DAPI, myofibers were permeabilized and blocked in 5% goat serum/2% BSA/0.2% Triton X-100/PBS for 10 min at room temperature, inverting. Next, myofibers were washed three times with PBS and mounted with VectaShield containing DAPI (Vector Laboratories). Myofibers were imaged using a Nikon SpectraX widefield microscope. Nuclei were counted and myofiber length was measured in 3D reconstructed images using Imaris software (Bitplane). Raw number of myonuclei per fiber and myofiber length was recorded, and these data were used to determine the average number of myonuclei per fiber in each condition. For Figs. 2 and 6, myofiber volume was determined by measuring fiber diameter at five spots along the length of the fiber, using the average diameter to calculate cross-sectional area of the fiber, and multiplying that by the length of each fiber.

For analysis in Fig. 5a–d, myofibers were stored in 1% PFA before being transferred to a petri dish with distilled water. Single fibers were placed on a glass slide (Superfrost Plus, Thermo Fisher Scientific) and the slides were mounted by a glass cover slip (No. 1.5, Marienfeld) with DAPI Fluoromount-G (Southern Biotech) to visualize nuclei. Slides were dried overnight and sealed with nail polish and stored no longer than 2 days before imaging. Images were acquired with a 40× oil immersion objective (CPI, Plan Fluor, NA 1.3) on an Andor DragonFly (Andor, Oxford Instruments) confocal microscope with a Zyla4.2 sCMOS camera with a x–y resolution of $0.3 \times 0.3$ μm$^2$ and z step size of 1 μm. Lasers with emission wavelength 405 nm (DAPI), 488 and 561 nm were used. Pixel binning of $2 \times 2$ was used to reduce time of image acquisition and improve signal-to-noise ratio. 25–40 cells from each muscle were analyzed. Imaged segments along the fibers were chosen visually from a straight section from the mid 1/3 of each fiber, excluding NMJ and MTJ regions. Nuclear number and cellular volume were analyzed using the Imaris Bitplane 9.20 software (Oxford Instruments). Using the spot function in Imaris, a spot was automatically assigned to each nucleus based on the fluorescence intensity from DAPI, followed by manual corrections. Volume and surface rendering were performed using the tissue autofluorescence in the red and green channel. Rendering of cellular geometry was performed using the fluorescence perimeter border of the cell in the cross-sectional direction as an outer limit, thereby preserving cell morphology during quantification.

**In situ muscle force measurements**. At 5 months of age, isometric force of the tibialis anterior (TA) muscle was measured in both control and Δ2w mice. Animals were anesthetized using isoflurane and kept warm under a radiant heating lamp throughout the experiment. An incision parallel to the femur was made to expose and cut the distal portion of the sciatic nerve close to the hip. The tibia branch of the sciatic nerve at the knee was also cut, allowing for stimulation of the peroneal nerve leading to the TA muscle. The distal TA tendon was severed, and the lower 1/3 of the muscle was released from the tibia. Blood circulation to the TA remained intact, and the portion of TA still connected to the tibia remained under the facia, which kept its temperature and moisture. The leg was immobilized at the knee using the limb clamp, and the foot was secured to the platform using surgical tape. A silk suture was tied to the free end of the TA tendon and attached to the lever arm of the apparatus (305C, Aurora Scientific), securing the TA muscle parallel to the platform to measure force production. Two electrodes were placed against the sciatic nerve for direct stimulation at 50 mA with 0.2 ms pulses, adjusting the lever arm to achieve optimal length of the TA muscle allowing for maximal isometric twitch force. Peak isometric tetanic force was measured by stimulating the sciatic nerve at 50 mA with a frequency of 150 Hz for 350 ms. Force output was collected by Dynamic Muscle Control (DMC v5.5) and analyzed by Dynamic Muscle Analysis (DMA v5.3) (Aurora Scientific). After the force measurements, tibialis anterior length and mass were measured to calculate physiological cross-sectional area (PCA). PCA was calculated using the following formula [muscle mass (g) / 1.06 (fiber density) × muscle length (cm) × 0.6 (muscle length to fiber length ratio)][53]. Specific force of individual muscles were determined by normalizing the maximal isometric tetanic force (Po) to PCA. To evaluate fatigue resistance, repeated isometric tetanic contractions were induced at 150 Hz every 10 s for a total of 50 contractions. The loss of force generation after every 10 contractions was calculated and plotted as a percentage of initial force.

**Single myofiber force production**. TA muscle samples, from 5 month-old mice, were placed in relaxing solution at 4 °C and treated with skinning solution (relaxing solution containing glycerol; 50:50 v/v) for 24 h at 4 °C. Bundles of ~50 myofibres were then dissected and transferred to −20 °C[54]. On the day of experiment, single myofibers were dissected from TA bundles in a relaxing solution. They were then individually attached between connectors leading to a force transducer and a lever arm system (model 1400 A; Aurora Scientific). Sarcomere length was set to ≈2.50 μm and the temperature to 15 °C. Fiber cross-sectional area (CSA) was estimated from the width and depth, assuming an elliptical circumference. The absolute maximal isometric force generation was calculated as the difference between the total tension in the activating solution (pCa 4.50) and the resting tension measured in the same myofiber while in the relaxing solution (pCa 9.0). Specific force was defined as absolute force divided by CSA. Relaxing and activating solutions contained 4 mM Mg-ATP, 1 mM free Mg$^{2+}$, 20 mM imidazole, 7 mM EGTA, 14.5 mM creatine phosphate, and KCl to adjust the ionic strength to 180 mM and pH to 7.0. The concentrations of free Ca$^{2+}$ were $10^{-9.0}$ M (relaxing solution) and $10^{-4.5}$ M (activating solution)[54].

**Inhibition of myostatin signaling**. ACVR2B-Fc serves as a decoy receptor for myostatin, which effectively inhibits myostatin/activin A signaling and induces robust muscle hypertrophy, as previously described[29,39]. At 3.5 months of age, both control and Δ2w mice were given ACVR2B-Fc by intraperitoneal injection at a dose of 10 mg/kg. This dose was administered once a week for 4 weeks before tissue was harvested for subsequent analysis.

**RNA analysis**. Flash-frozen muscle samples were weighed, and total RNA extracted using standard Trizol-based protocols. Quality and concentrations of RNA were determined using the Agilent 2100 Bioanalyzer [Gene Expression Core (GEC) Facility (CCHMC)]. Microarray analysis for differential gene expression (DGE) profiles was carried out at the GEC facility using the Affymetrix Clariom S platform. Bioinformatics analyses of resultant data (CEL files) to determine DGE between samples was carried out using the Transcriptome Analysis Console (Applied Biosystems; ver. 4.0.0.25), the Clariom_S_Mouse TAC Configuration file (ver. 2), and the iPathwayGuide (Advaita Bioinformatics). Gene Ontology enrichment analysis for categorization by biological processes was achieved utilizing the Panther Classification System[55,56]. Heat maps were generated from gene-signals generated utilizing the Signal Space Transformation Robust Multi-Array Average normalization method (sst-rma-gene-signals) for each gene of interest using the Heatmapper software[57]. All original microarray data were deposited in the NCBI's Gene Expression Omnibus database (GSE141296).

mRNA contributions to total RNA content[3] were calculated as a metric of mRNA concentration (used as a measurement of adaptability in the genetic models where myonuclear numbers were titrated) by normalizing sst-rma-gene signals for every gene to total RNA amounts of the same sample. This value was then normalized to EDL nuclei numbers to yield mRNA concentration on a per nuclear basis. First-strand cDNA synthesis was carried out using the Superscript III Kit (Invitrogen 18080400) using oligo dT primers. Semi-quantitative RT-PCR analysis was performed on a C1000 Touch CFX96 real-time PCR machine (Bio-Rad) with the following primers: Myh1: forward, 5′-CGGAGTCAGGTGAATACTCACG-3′; reverse, 5′-GAGCATGAGCTAAGGCACTCT-3′, Myh4: forward, 5′-CTTTGCT TACGTCAGTCAAGGT-3′; reverse, 5′-AGCGCCTGTGAGCTTGTAAA-3′, Tnnc2: forward, 5′-GAGGCCAGGTCCTACCTCAG-3′; reverse, 5′-GGTGCCCAA CTCTTTAACGCT-3′, Tnnt3: forward, 5′-GGAACGCCAGAACAGATTGG-3′; reverse, 5′-TGGAGGACAGAGCCTTTTTCTT-3′, Acta1: forward, 5′-CCCAAAG CTAACCGGGAGAAG-3′; reverse, 5′-CCAGAATCCAACACGATGCC-3′. Results were normalized to glyceraldehyde phosphate dehydrogenase (GAPDH) using the following primers: forward, 5′-TGCGACTTCAACAGCAACTC-3′; reverse, 5′- GCCTCTCTTGCTCAGTGTCC-3′.

Measures of transcript abundance (used when not assessing myonuclear flexibility) for Myh1, Myh4, Tnnc2, Tnnt3, and Acta1 were generated as a product of expression levels determined by semi-quantitative RT-PCR as described above for each sample and the total RNA amounts of the same sample. Transcript abundance on a per nuclear basis was subsequently calculated by normalizing that product to average myonuclear number (from the EDL).

**Protein analysis**. After recording of weights, flash-frozen muscle samples were pulverized in a liquid nitrogen-cooled mortar and pestle, and then mechanically homogenized on ice in known volumes of homogenization buffer [20 mM Tris (pH 7.3 with 1 N HCl), 1 mM EDTA, 1 mM dithiothreitol (DTT), 0.5% Triton X-100 + protease and phosphatase inhibitor cocktails (Sigma, P8340, P5726)]. Solubilization was allowed to proceed on a nutator for 2 h at 4 °C, after which protein concentrations were measured using the Coomassie Plus (Bradford) Protein Assay Kit (ThermoScientific) against a BSA-based standard curve. Protein lysates were prepared for SDS-PAGE analysis by heating at 95 °C for 10 min in 1× Laemmli sample buffer containing 100 mM DTT. Proteins (2.5 μg/lane) were resolved on discontinuous polyacrylamide gels (7.5% for myosin heavy chains and 11.25% for sarcomeric actin), and transferred to 0.2 μm nitrocellulose membranes (BioRad, 162-0112). Membranes blocked in 5% BSA were incubated with respective primary antibodies diluted in 5% BSA [MF20 (1:20) deposited in DHSB[58] against Myh

isoforms, A2172 (1:1000; Sigma) against sarcomeric actin, and 10R-G109A (1:10000; Fitzgerald) against GAPDH]. The resulting immunoblots generated after incubation with relevant secondary antibodies [LiCOr Donkey anti-mouse IgG against MF20 (926-68072), Goat anti-mouse IgM against A2172 (926-68080), and Goat anti-mouse IgG against 10R-G109A (926-32210)] were scanned, imaged and analyzed for densitometry using the Odyssey CLx imaging system (LI-COR Biosciences, 9140).

**Statistical analysis**. Data were processed and analyzed using Microsoft Excel and GraphPad Prism 8 software. Data sets are presented as mean ± SD, and data were compared between groups using one-way ANOVA with a Tukey correction for multiple comparisons, two-way ANOVA with a Tukey correction for multiple comparisons, linear regression, or unpaired t-test. If unequal variances were observed, a Welch's t-test was used. Normality of data was also analyzed and if non-normal, Mann–Whitney tests were used for comparison between two groups or one-way ANOVA with a Kruskal–Wallis correction for multiple comparisons was used. Specific statistical tests used are notated in individual figure legends. Statistical significance was determined as $p < 0.05$.

**Reporting Summary**. Further information on research design is available in the Nature Research Reporting Summary linked to this article.

## Data availability
The datasets generated during the current study are available in the NCBI Gene Expression Omnibus database (GSE141296), or are available from the corresponding author upon reasonable request. Source data are provided with this paper.

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

## Acknowledgements

We thank Dr. Se-Jin Lee (Jackson Laboratory) for the ACVR2B-Fc reagent. This work was supported by grants to D.P.M. from the Cincinnati Children's Hospital Research Foundation, National Institutes of Health (R01AR068286, R01AG059605), and Pew Charitable Trusts. E.E. was funded by the US-Norway Fulbright Foundation for Educational Exchange. Work in the K.G. laboratory was funded by a grant from the Norwegian Research Council (grant: 240374). J.O. is funded by the Medical Research Council of the UK (MR/S023593/1). S.S. has received support from National Institutes of Health grants R01 HL130356, R56 HL139680, R01 AR067279, R01 HL105826 and R01 HL143490, R13 HL149313; American Heart Association 2019 Institutional Undergraduate Student (19UFEL34380251) and transformation (19TPA34830084) awards; and MyoKardia, AstraZeneca, Merck and Amgen. T.S. is supported by an American Heart Association Postdoctoral Fellowhip (19POST34380448).

## Author contributions

A.A.W.C., V.P., E.E., K.-A.H., K.G., and D.P.M. designed experiments. A.A.W.C., V.P., E.E., T.S., K.-A.H., and D.P.M. performed experiments. All authors analyzed the data and contributed to interpretations. D.P.M. wrote the manuscript with assistance from all authors.

## Competing interests

The authors declare no competing interests.
