## [Peer Review File · Nature Communications]

Reviewers' Comments:

Reviewer #1:

Remarks to the Author:

This study investigates the effect of the number of nuclei, which are regulated by the number of fusion events, on muscle fiber growth. Genetic inhibition of cell fusion at specific post-natal timepoints resulted in reduced nuclear numbers when compared to controls. An early fusion block (d1w) resulted in 75% reduction of nuclei, which was not enough for establishing functioning muscle fibers and resulted in early organismal death. A later fusion block (d2w) resulted in myofibers with a ~45% reduction of nuclei; however the mice survived normally, albeit with smaller muscle fibers. When normalized for muscle size, these fibers had a specific force comparable to controls. The analyses of mRNA data—presented as amount per nucleus under different conditions-- were striking and indicated that d2w condition, with ~45% less nuclei, resulted in functioning muscle fibers in which nuclei compensated by increasing mRNA levels. However, a 25% reduction of nuclei, by preventing nuclear accretion at a later timepoint (d3w), did not initiate this compensatory upregulation in gene expression. Strikingly, d2w and d3w muscles ended up with similar sizes through different nuclear scaling relationships.

Overall, the data are convincing; as well, the writing and authors' logic are very clear. This is a thoughtfully crafted manuscript and a pleasure to read.

Comments/critiques:

1. Multiple measurements for muscle size (cross sections, volumes) were given.

This is superb and provides a more comprehensive picture for the reader; however, an explanation of how the different measurements relate to each other, in addition to their scaling with nuclear number, would help. Also, muscle surface areas could be calculated and provide an additional size parameter.

2. Different Muscles (TA, EDL,...) and muscle fiber types: Different muscles are used in the different figures. These consist of a majority of fast fibers; do the data hold for slow muscle fibers such as the soleus? Are there similarities or differences of growth patterns and myonuclear domain changes?

3. It isn't clear from the methods whether the authors looked at both sexes. This is a parameter that should be considered in these studies.

4. Under the conditions in which the muscles are smaller (d2w, d3w), the muscle fibers are smaller. The authors show a representative image of the mice—but this is qualitative (Fig 3F). The muscle functions as a myocrine organ that could influence bone and other organs. Some measure of the total animal, be it the overall size, weight, hindlimb lengths should be considered: the question is : are the muscles scaling to the size of the organism?

5. The data presented in this study do not conclusively prove that it is nuclear number that directly determines whether a cell is able to flexibly regulate its synthetic output. Yes there is a strong correlation, but there are other factors that could be involved. It would be interesting take a closer look at the correlation of size (e.g. cell volume) and nuclear number for individual cells at each timepoint. Changes in the slope of a regression line on a log plot could indicate whether bigger or smaller cells are more affected by preventing additional fusions.

6. Size of muscle nuclei: many studies have shown that nuclear size is an indicator for activity in diploid cells. The images in Fig.1 suggest that the size of the nuclei is increased in d1w and d2w fibers compared to controls. Measurement of nuclear sizes under different conditions could improve the impact of this study.

7. 3D reconstructions of fiber segments (Figure 5a): these were difficult to interpret for several

reasons: how was the position along the fiber chosen? Was NMJ or MTJ location considered?

8. Muscle function: Are there any changes in dw3 mice? These data would be good to have to complete the study. Also, could the authors comment on the general state of movement of the dw1 and dw2 mice—do they run on a wheel, stay in corners, show wildtype behaviors?

9. The authors argue that flexibility is 'regulated by the number of nuclei'. The fusion block in d2w and d3w muscles occurred at different times/developmental timepoints. Is it possible that the flexibility of the muscle nuclei just changes with developmental time? Or that different growth programs operate early and late? What if there is a deficit threshold during development which regulates whether compensation is activated or not? This is also relevant in the Discussion: the analysis of the mRNA output per nucleus in a denervation model might provide some insight to this issue.

Additional specific comments:

Line150: 'not a strict linear relationship': A series of box plots does not support this conclusion. Comparison of different parameters in scatter plots showing individual measurements could reveal relationships that could exist in control muscles, and could be altered under conditions of fusion block.

Line 157. 'heirarchachal' should be spelled 'hierarchical'

Line 172 'kyphosis' is spelled correctly here but not in Figure 3 legend.

Line179: 'nuclei are normal and lack any genetic perturbation': where is this shown?

Line189+: The difference between mRNA levels in d1w and d2w muscles might be easier to see if data in Fig 4c and d would be combined into one plot. Same for e and f.

Line208+: the clarity of the questions could be improved

Line215: fiber segments: how were nuclei/domains at the edges of the segments quantified?

Line238: Graph in Fig 5g indicates increased protein content in d2w muscles – text states similar levels. Please clarify.

Line 473: Figure 4a. There seems to be a bimodal distribution in the control (total RNA/muscle weight). Could the authors comment on this?

Statistical analysis is appropriate.

Reviewer #2:

Remarks to the Author:

Cramer and co-workers have investigated the role of myonuclear accretion muscle size by reducing myonuclei number during neonatal development. It is stated that: 1) nuclei number determine size and muscle function, and 2) there is a negative relationship between myonuclei number and elevations in mRNA concentrations once a critical number of myonuclei has been obtained.

General comments: This is an interesting study where the authors aim to improve our understanding of the regulation of muscle fiber size by myonuclei number. The overall enthusiasm is, however, hampered by the bold statements related to regulation of muscle size and function based on mRNA content and in vivo muscle function measurements during postnatal development (see below). Other statements such as: there is not a strict linear relationship between muscle size and myonuclear number or that there is a myonuclear domain size threshold for maintaining muscle size/function are not novel.

Regulation of muscle size by accretion of myonuclei is largely based a relationship between size and mRNA concentration or a lack thereof ($\Delta 3w$). Muscle fiber size is primarily related to the balance between protein synthesis and degradation, and there are multiple factors which may influence net protein content besides mRNA levels. There is accordingly a risk the authors are overestimating the importance of mRNA content in the regulation of muscle size. Primary mechanisms may rely on translational factors, protein transport and assembly only weakly related to total mRNA levels. The authors also discuss the potential role of differences in mRNA production and degradation may have an impact on the measured mRNA levels. This is a valid comment and additional uncertainty to statements based on mRNA content in the different groups.

The authors use their specific force measurements as an indicator of “functionally normal muscle”. However, the method to measure specific force used in this study has methodological limitations and the calculation of “physiological cross-sectional area” is based on assumptions of constant fiber density and muscle length to fiber length ratio. However, the TA has a complex fiber orientation and it cannot be assumed that these characteristics are identical in the different groups investigated. In fact, the shorter muscle fibers in response to reduced number of myonuclei indicate a change in muscle architecture and the authors also suggest a change in pennation angle as a mechanism underlying the lack of contractures. In addition, there are fiber type specific differences in specific force and there are myosin isoform transitions during development which have not been investigated in the different groups studied in the current project. Some of these limitations are mentioned in the Discussion, but this does not improve the interpretative value of the calculated specific forces.

Specific comments:

It is suggested that the authors also include the body weights of the different groups of mice including WT so the effects of muscle size on overall mouse weight can be evaluated.

p.10 l. 226-227 It has been confirmed in different studies that there is a significant increase in muscle size without a proportional increase in myonuclear number in myostatin knock out mice, supporting the current observations and the authors may consider including this in the Discussion.

Specify in more detail how muscle force was measured. There is no information on stimulus strength or muscle temperature monitoring (keeping the mouse warm with a lamp is not sufficient).

Fig. 5g According to the graph, there is an increased amount of contractile proteins normalized to muscle weight in $\Delta 2w$ mice. This could indicate an increased force generation capacity $\Delta 2w$ mice

unless there is an increased amount of contractile proteins with a decreased functional capacity in the $\Delta 2w$ mice such as a higher content of developmental isoforms, alternatively a suboptimal incorporation of contractile proteins in the sarcomere. Please comment.

Minor comments:

p.12 l. 271 please rephrase this sentence.

p.35 this data was..... change to these data were.....

p.36 l.699 It is stated that the average diameter was used to calculate the cross-sectional area. However, it is obvious from Fig. 5 that the fibers do not have a circular cross-section and in the graph it is stated that myofiber area is based on the confocal images. Please also add information on how nuclei cut at the ends of the fiber segments were treated in the calculation of myonuclear domains.

We thank the reviewers for their efforts to improve our manuscript. We have responded to each suggestion with either new data or additional text. Our response to each reviewer critique is below in red font.

Reviewers' comments:

Reviewer #1 (Remarks to the Author):

This study investigates the effect of the number of nuclei, which are regulated by the number of fusion events, on muscle fiber growth. Genetic inhibition of cell fusion at specific post-natal timepoints resulted in reduced nuclear numbers when compared to controls. An early fusion block (d1w) resulted in 75% reduction of nuclei, which was not enough for establishing functioning muscle fibers and resulted in early organismal death. A later fusion block (d2w) resulted in myofibers with a ~45% reduction of nuclei; however the mice survived normally, albeit with smaller muscle fibers. When normalized for muscle size, these fibers had a specific force comparable to controls. The analyses of mRNA data—presented as amount per nucleus under different conditions-- were striking and indicated that d2w condition, with ~45% less nuclei, resulted in functioning muscle fibers in which nuclei compensated by increasing mRNA levels. However, a 25% reduction of nuclei, by preventing nuclear accretion at a later timepoint (d3w), did not initiate this compensatory upregulation in gene expression. Strikingly, d2w and d3w muscles ended up with similar sizes through different nuclear scaling relationships.

Overall, the data are convincing; as well, the writing and authors' logic are very clear. This is a thoughtfully crafted manuscript and a pleasure to read.

Comments/critiques:

1. Multiple measurements for muscle size (cross sections, volumes) were given. This is superb and provides a more comprehensive picture for the reader; however, an explanation of how the different measurements relate to each other, in addition to their scaling with nuclear number, would help. Also, muscle surface areas could be calculated and provide an additional size parameter.

The reviewer highlights an extremely interesting concept about the potential role of muscle surface area (or surface domain dimensions) in determining myofiber volume, and the mechanistic relationship between that parameter and myonuclear transcriptional output. We believe investigating these novel questions merits an independent, in-depth study. We are a little uncertain about how to incorporate into the manuscript an explanation about how the different measurements (which we assume are CSA and volume) relate to each other, but it is clear that nuclei numbers may have a greater influence on certain aspects of muscle including width, length, volume and surface area, which is highlighted in the discussion. At this time, the accompanying paper from the Gundersen laboratory reports on the scaling relationship between nuclear numbers and surface area domains.

2. Different Muscles (TA, EDL,...) and muscle fiber types: Different muscles are used in the different figures. These consist of a majority of fast fibers; do the data hold for slow muscle fibers such as the soleus? Are there similarities or differences of growth patterns and myonuclear domain changes?

This is an excellent question and we agree that including some analysis of the soleus would be more comprehensive. We did analyze single myofibers from soleus muscles of control and $\Delta 2w$ mice at P42 (n=3 per group, 20 myofibers/mouse). Although the results are interesting and consistent with our interpretations, we do not think it would be justified to report any definitive conclusions based on the below preliminary findings.

First, there is more dramatic reduction (70%) of nuclei number in the soleus of $\Delta 2w$ mice compared to the fast muscles (55% reduction of nuclei). A reason for this could be due to different kinetics of muscle development, or that oxidative myofibers are associated with persistent low-level of myonuclear accrual. We observed a 9% reduction in length and 33% reduction in volume, which is similar to the reductions in the $\Delta 2w$ EDL. Overall, those data suggest a potentially more dramatic adaptive response in terms of establishment of size in the soleus but again this requires much more analysis. No doubt that future work will be needed to understand the effects of nuclear reduction on the metabolic demands of slow myofibers, but we think these results do not impact the claims in the current paper regarding the ability of developing myofibers to adapt to reduced myonuclear numbers in establishment of size and volume. We added the clarification in the first line of the discussion that this work is specifically relevant to fast skeletal muscle.”

3. It isn't clear from the methods whether the authors looked at both sexes. This is a parameter that should be considered in these studies.

We completely agree gender should be considered and apologize for not including this in the methods of the original submission. In the revised version we more specifically describe that both males and females were used. All comparative groups contained the same percentage of males and females. In some cases, we analyzed males and females separately and results were consistent. We have mentioned that equal ratios of males and females were analyzed in the methods.

4. Under the conditions in which the muscles are smaller (d2w, d3w), the muscle fibers are smaller. The authors show a representative image of the mice—but this is qualitative (Fig 3F). The muscle functions as a myocrine organ that could influence bone and other organs. Some measure of the total animal, be it the overall size, weight, hindlimb lengths should be considered: the question is : are the muscles scaling to the size of the organism?

This is a great point and we now show body weight and tibia lengths for the genetically modified mice. Overall, these data show that growth and establishment of tibia length were unaltered.

5. The data presented in this study do not conclusively prove that it is nuclear number that directly determines whether a cell is able to flexibly regulate its synthetic output. Yes there is a strong correlation, but there are other factors that could be involved. It would be interesting take a closer look at the correlation of size (e.g. cell volume) and nuclear number for individual cells at each timepoint. Changes in the slope of a regression line on a log plot could indicate whether bigger or smaller cells are more affected by preventing additional fusions.

This is an excellent point from the reviewer. We do want to note that the graph suggested here (correlation of size and volume) was shown in the accompanying scaling paper by the Gundersen laboratory.

We also agree that it is difficult to definitively prove within the scope of this manuscript that nuclei number controls flexibility. However, we have considered other possibilities including that volume controls nuclear output, which has been proposed for mononuclear cell types. We tested if increases in size, without addition of new nuclei, would regulate synthetic transcriptional output on a per nuclear basis. To achieve this, we treated control and $\Delta 2w$ mice with ACVR2B-Fc, which acts as a decoy receptor for myostatin, and induces dramatic increases in muscle size. It is known that this stimulus does not elicit increases in myonuclear accretion. This analysis showed that mRNA concentrations are not increased, even in WT mice, indicating that an increase in myofiber volume is not an obligatory driver of mRNA concentrations. This is consistent with our interpretations that nuclear number and associated mRNA concentrations are more likely to be the main determinants of size and flexibility.

6. Size of muscle nuclei: many studies have shown that nuclear size is an indicator for activity in diploid cells. The images in Fig.1 suggest that the size of the nuclei is increased in d1w and d2w fibers compared to controls. Measurement of nuclear sizes under different conditions could improve the impact of this study.

The DAPI images shown in Fig. 1 are not of sufficient resolution to make precise measurements of nuclear size. The revised version of the accompanying paper from the Gundersen laboratory measured nuclei size in control and $\Delta 2w$ 3D-rendered myofibers and did not observe any differences.

7. 3D reconstructions of fiber segments (Figure 5a): these were difficult to interpret for several reasons: how was the position along the fiber chosen? Was NMJ or MTJ location considered?

The NMJ and MTJ were not considered in this analysis. We think it is an outstanding question about how those areas of the muscle are controlled. However, segment along the fibers was chosen visually from a straight section from the mid 1/3, and the NMJ and MTJ were visually excluded from imaging. That detail has been added to the methods.

8. Muscle function: Are there any changes in dw3 mice? These data would be good to have to complete the study. Also, could the authors comment on the general state of movement of the dw1 and dw2 mice—do they run on a wheel, stay in corners, show wildtype behaviors?

As function when normalized to size was unaltered in $\Delta 2w$ mice, we do not anticipate the $\Delta 3w$ model to behave differently; therefore, while functional analysis of the $\Delta 3w$ model would undoubtedly provide additional information, we are not certain results would change or advance the interpretations.

Also, as described below, an important point to highlight (as we have in the revised version of the manuscript) is that the adaptive response is not absent in the $\Delta 3w$ model – rather our data reveal that the magnitude of that response on a per nuclear basis (or in other words, myonuclear flexibility) correlates negatively with myonuclear numbers, and is therefore much milder in the $\Delta 3w$ model.

We did not observe any behavioral differences in $\Delta 2w$ mice, and putting them through any exercise regimen that might require myonuclear accretion is problematic given that the genetic lesion causing an inability to fuse is permanent, a condition we have shown to result in exercise intolerance and maladaptive responses (Goh et al. *eLife* 2019). In terms of the $\Delta 1w$ mice, they move around the cage but as they age and get closer to death, they seem to become less motile.

9. The authors argue that flexibility is ‘regulated by the number of nuclei’. The fusion block in d2w and d3w muscles occurred at different times/developmental timepoints. Is it possible that the flexibility of the muscle nuclei just changes with developmental time? Or that different growth programs operate early and late? What if there is a deficit threshold during development which regulates whether compensation is activated or not? This is also relevant in the Discussion: the analysis of the mRNA output per nucleus in a denervation model might provide some insight to this issue.

This is another great question by the reviewer. To further test the idea that flexibility is regulated by nuclei number we more deeply analyzed $\Delta 3w$ mice. Through analysis of more animals we discovered some flexibility, in terms of mRNA output on a per nuclear

basis. Specifically, we performed qPCR for sarcomeric genes (Figure 7e). These data suggest that flexibility is not encoded based on developmental time and provide further evidence that flexibility is regulated in some fashion by the number of nuclei.

Additional specific comments:

Line150: 'not a strict linear relationship': A series of box plots does not support this conclusion. Comparison of different parameters in scatter plots showing individual measurements could reveal relationships that could exist in control muscles, and could be altered under conditions of fusion block.

We replaced 'not a strict linear relationship' in the revised version.

Line 157. 'heirarchachal' should be spelled 'hierarchical'

This change has been made. Thanks for catching the mistake.

Line 172 'kyphosis' is spelled correctly here but not in Figure 3 legend.

We corrected the typo.

Line179: 'nuclei are normal and lack any genetic perturbation': where is this shown?

The reviewer is correct that we did not show this directly, but we want to highlight that there is no reason to think that the nuclei within a myofiber are genetically abnormal. The Cre driver is Pax7-driven, therefore no genetic recombination is anticipated to occur in myonuclei.

Line189+: The difference between mRNA levels in d1w and d2w muscles might be easier to see if data in Fig 4c and d would be combined into one plot. Same for e and f.

This is an excellent idea and has been incorporated into the revised version.

Line208+: the clarity of the questions could be improved

We attempted to clarify and simplify these questions.

Line215: fiber segments: how were nuclei/domains at the edges of the segments quantified?

A segment along the fibers was chosen visually from a straight section from the mid 1/3, and the NMJ and MTJ were visually excluded from imaging.

Line238: Graph in Fig 5g indicates increased protein content in d2w muscles – text states similar levels. Please clarify.

This has been edited to clarify that overall protein levels are similar but are increased in $\Delta 2w$ mice when normalized to muscle mass.

Line 473: Figure 4a. There seems to be a bimodal distribution in the control (total RNA/muscle weight). Could the authors comment on this?

This is an interesting observation, but we think that any comment would be too speculative. It is possible that this issue is due to variability between litters.

Statistical analysis is appropriate.

Reviewer #2 (Remarks to the Author):

See uploaded pdf file

Cramer and co-workers have investigated the role of myonuclear accretion muscle size by reducing myonuclei number during neonatal development. It is stated that: 1) nuclei number determine size and muscle function, and 2) there is a negative relationship between myonuclei number and elevations in mRNA concentrations once a critical number of myonuclei has been obtained.

General comments: This is an interesting study where the authors aim to improve our understanding of the regulation of muscle fiber size by myonuclei number. The overall enthusiasm is, however, hampered by the bold statements related to regulation of muscle size and function based on mRNA content and in vivo muscle function measurements during postnatal development (see below). Other statements such as: there is not a strict linear relationship between muscle size and myonuclear number or that there is a myonuclear domain size threshold for maintaining muscle size/function are not novel.

We are thankful that the reviewer finds our manuscript interesting. We have dealt with each of the major concerns below. We have edited our statement that 'there is not a strict linear relationship between muscle size and myonuclear number'. It is difficult for us to comment on the reviewer's point that some of our observations are not novel since no explanation or citations are given.

To our knowledge, there is no previous study that

(a) reports the ability to titrate myonuclei numbers during post-natal development;
(b) utilizes this unique experimental intervention to interrogate, for the first time, the requirement of myonuclear accrual in establishment of myofiber size, volume, and

function; and,

(c) provides evidence of an inverse relationship between mRNA concentrations and nuclear numbers in a syncytial tissue.

We anticipate that the revisions made will help to better highlight the focus of this study, which are the factors that contribute to, and determine, the establishment of myonuclear domain size during post-natal development, rather than the relationship between myonuclear domain size threshold(s) and maintenance of muscle size/function.”

Regulation of muscle size by accretion of myonuclei is largely based a relationship between size and mRNA concentration or a lack thereof ($\Delta 3w$). Muscle fiber size is primarily related to the balance between protein synthesis and degradation, and there are multiple factors which may influence net protein content besides mRNA levels. There is accordingly a risk the authors are overestimating the importance of mRNA content in the regulation of muscle size. Primary mechanisms may rely on translational factors, protein transport and assembly only weakly related to total mRNA levels. The authors also discuss the potential role of differences in mRNA production and degradation may have an impact on the measured mRNA levels. This is a valid comment and additional uncertainty to statements based on mRNA content in the different groups.

This is a great point that a major regulator of muscle size is protein synthesis and degradation. We would like to emphasize that our analyses and interpretations do not discount the role of proteostasis in muscle. Indeed, we agree that there are likely multiple factors that influence protein content, and contribute to muscle size and function. Our study investigates, and highlights, the role mRNA content plays in the establishment of myonuclear domains and muscle size during post-natal development. Our reasoning for this focus is that by titrating nuclear accrual, myonuclear numbers are the main perturbation in our system, which directly impacts the templates available for protein synthesis and degradation. Nonetheless, we have expanded our discussion to include how proteostasis may fit into our models.

The authors use their specific force measurements as an indicator of “functionally normal muscle”. However, the method to measure specific force used in this study has methodological limitations and the calculation of “physiological cross-sectional area” is based on assumptions of constant fiber density and muscle length to fiber length ratio. However, the TA has a complex fiber orientation and it cannot be assumed that these characteristics are identical in the different groups investigated. In fact, the shorter muscle fibers in response to reduced number of myonuclei indicate a change in muscle architecture and the authors also suggest a change in pennation angle as a mechanism underlying the lack of contractures.

To complement our functional analyses on the TA, our collaborator (Dr. Julien Ochala) has performed single fiber force measurements. This analysis also shows no effect on specific force in $\Delta 2w$ mice (Figure 5g) consistent with the idea that these mice with

fewer myonuclei are able to adapt to generate functionally normal sarcomeres and muscle.

In addition, there are fiber type specific differences in specific force and there are myosin isoform transitions during development which have not been investigated in the different groups studied in the current project. Some of these limitations are mentioned in the Discussion, but this does not improve the interpretative value of the calculated specific forces.

In the revised version of the manuscript, we analyze fiber type in the $\Delta 2w$ TA and observed normal fiber type distributions (Supplementary Figure 4). These data provide evidence that specific force in $\Delta 2w$ mice is not compensated for by fiber type change.

Specific comments:

It is suggested that the authors also include the body weights of the different groups of mice including WT so the effects of muscle size on overall mouse weight can be evaluated.

We have included these data in the revised version.

p.10 l. 226-227 It has been confirmed in different studies that there is a significant increase in muscle size without a proportional increase in myonuclear number in myostatin knock out mice, supporting the current observations and the authors may consider including this in the Discussion.

There is indeed an increase in muscle size without an increase of myonuclear number in myostatin KO mice. However, these mice are very different from the $\Delta 2w$ mice we report here as the myostatin KO mice do not exhibit increased specific force (discussed and referenced in the revised version of the manuscript). It has been reported that while myostatin KO mice have increased muscle size, the increase in volume is not a direct consequence of increasing the number of myofibrils in parallel. Perhaps the reviewer is using the myostatin KO mouse to suggest that our results with the $\Delta 2w$ mice are not novel. We would respectfully disagree with the argument that $\Delta 2w$ mice are similar to myostatin KO mice because the specific force in $\Delta 2w$ mice is normal. Additionally, we treated control and $\Delta 2w$ mice with a myostatin decoy receptor (ACVR2B-Fc) and found that while muscle size increased in both groups of mice, there was not an increase in mRNA concentrations after ACVR2B-Fc treatment. This further highlights the potential importance of mRNA content in the regulation of functional muscle size. We have discussed these points in the revised version of the manuscript.

Specify in more detail how muscle force was measured. There is no information on stimulus strength or muscle temperature monitoring (keeping the mouse warm with a lamp is not sufficient).

We apologize for the lack of detail here. We stimulated the sciatic nerve at 50mA with frequency of 150Hz for 350ms to measure peak isometric tetanic force. We tried to keep the body temperature stable with heating lamp because we are unable to measure TA muscle temperature directly. The proximal tendon of the TA was still attached to the knee and blood circulation was intact during the measurement so if body temperature is stable, we can expect the muscle is also stable in temperature. Furthermore, as we mentioned in the methods, we only isolate the lower 1/3 of the TA from tibia and kept the rest of the TA under the fascia, which covered the TA to keep its temperature and moisture. These details have been added to the methods.

Fig. 5g According to the graph, there is an increased amount of contractile proteins normalized to muscle weight in $\Delta 2w$ mice. This could indicate an increased force generation capacity $\Delta 2w$ mice unless there is an increased amount of contractile proteins with a decreased functional capacity in the $\Delta 2w$ mice such as a higher content of developmental isoforms, alternatively a suboptimal incorporation of contractile proteins in the sarcomere. Please comment.

Adaptive increase in volume & mass of skeletal muscle, as in the case of myostatin inhibition, can be non-sarcomeric where the myofibrillar protein content is not preserved at normal levels. We carried out immunoblot analyses for the two main contractile proteins (MYH and S. ACTIN) to determine if the growth observed in $\Delta 2w$ muscle was sarcomeric. We now show the data for protein levels normalized to GAPDH and observed similar levels between control and $\Delta 2w$. We agree with the reviewer that it is difficult to interpret the original data normalized to weight of the muscle and have decided to not include that in the manuscript.

Minor comments:

p.12 l. 271 please rephrase this sentence.

This sentence has been rephrased.

p.35 this data was..... change to these data were.....

This has been changed.

p.36 l.699 It is stated that the average diameter was used to calculate the cross-sectional area. However, it is obvious from Fig. 5 that the fibers do not have a circular cross-section and in the graph it is stated that myofiber area is based on the confocal images. Please also add information on how nuclei cut at the ends of the fiber segments were treated in the calculation of myonuclear domains.

For Fig. 5 we did not use the average diameter to calculate CSA because these were 3D-rendered images. However, we did use average diameter for CSA calculations in

Fig. 2 and 6. This is now mentioned in the methods. Nuclei with 50% or more of their total size, based on neighboring nuclear size within the image, were included in the analysis.

Reviewers' Comments:

Reviewer #1:

Remarks to the Author:

This is a re-review of a manuscript from Cramer et al.

The authors were very receptive to the first round of comments; the changes in the text plus the additional experiments strengthen the manuscript and address all my concerns.

Minor corrections/suggestion:

Comment/suggestion for discussion: are there any physiological events in the mouse (hormones, innervation pattern ?) that occur during postnatal day 6 and 13 that may influence the ability of myonuclei/muscle to adapt vs not? That is, it is not the number per say but the identity of the nuclei that are being added at those different time points?

Line 364. Nuclie should be nuclei

line 499 annonated should be annotated

Supp Fig 2b P42—Y-axis is different from other plots, please check

Reviewer #2:

Remarks to the Author:

The authors have addressed my concerns satisfactory.

Red font is the authors' response.

Reviewer #1 (Remarks to the Author):

This is a re-review of a manuscript from Cramer et al.

The authors were very receptive to the first round of comments; the changes in the text plus the additional experiments strengthen the manuscript and address all my concerns.

Minor corrections/suggestion:

Comment/suggestion for discussion: are there any physiological events in the mouse (hormones, innervation pattern ?) that occur during postnatal day 6 and 13 that may influence the ability of myonuclei/muscle to adapt vs not? That is, it is not the number persay but the identity of the nuclei that are being added at those different time points?

This is an excellent point and we agree we can't be sure that the nuclei are different at these time points. Our discussion is already quite dense and we think adding another paragraph of speculation would be too much.

Line 364. Nuclie should be nuclei

This has been edited

line 499 annonated should be annotated

This has been edited

Supp Fig 2b P42—Y-axis is different from other plots, please check

The y-axis is correct here. It is different because it goes below 0.

Reviewer #2 (Remarks to the Author):

The authors have addressed my concerns satisfactory.